# THINK THEN EMBED: GENERATIVE CONTEXT IMPROVES MULTIMODAL EMBEDDING

**Xuanming Cui**[1][†] **Jianpeng Chen**[2][*] **Hong-You Chen**[2] **Satya Narayan Shukla**[2]
**Abhijeet Awasthi**[2] **Xichen Pan**[3] **Chaitanya Ahuja**[2] **Shlok Kumar Mishra**[2]
**Taipeng Tian**[2] **Qi Guo**[2] **Ser-Nam Lim**[1] **Aashu Singh**[2] **Xiangjun Fan**[2]
[1]University of Central Florida
[2]Meta
[3]New York University

## ABSTRACT

There is a growing interest in Universal Multimodal Embeddings (UME), where models are required to generate task-specific representations. While recent studies show that Multimodal Large Language Models (MLLMs) perform well on such tasks, they treat MLLMs solely as encoders, overlooking their generative capacity. However, such an encoding paradigm becomes less effective as instructions become more complex and require compositional reasoning. Inspired by the proven effectiveness of chain-of-thought reasoning, we propose a general Think-Then-Embed (TTE) framework for UME, composed of a reasoner and an embedder. The reasoner MLLM first generates reasoning traces that explain complex queries, followed by an embedder that produces representations conditioned on both the original query and the intermediate reasoning. This explicit reasoning step enables more nuanced understanding of complex multimodal instructions. Our contributions are threefold. *First*, by leveraging a powerful MLLM reasoner, we achieve state-of-the-art performance on the MMEB-V2 benchmark, surpassing proprietary models trained on massive in-house datasets. *Second*, to reduce the dependency on large MLLM reasoners, we finetune a smaller MLLM reasoner using high-quality embedding-centric reasoning traces, achieving the best performance among open-source models with a 7% absolute gain over recently proposed models. *Third*, we investigate strategies for integrating the reasoner and embedder into a unified model for improved efficiency without sacrificing performance.

## 1 INTRODUCTION

Multimodal embedding-based retrieval has emerged as a popular and effective solution for handling diverse data types such as text, images, and videos (Radford et al., 2021; Li et al., 2023a; Yu et al., 2022; Li et al., 2023b). Traditionally, these models focus on learning general-purpose representations that capture content similarity across modalities. Recently, there has been growing interest in instruction-aware Universal Multimodal Embeddings (UME) (Jiang et al., 2024; Gu et al., 2025). UMEs are powerful representations because they bridge the gap between general-purpose content similarity and user-specific requirements. This capability unlocks a wide range of applications, such as retrieving documents or images based on nuanced queries, powering personalized recommendation systems, supporting multimodal search with complex transformations, and tackling knowledge-intensive tasks where the same input may yield different embeddings depending on the downstream objective. To advance this direction, recent benchmarks like MMEB (Meng et al., 2025) aggregate diverse, instruction-driven retrieval tasks—including VQA, grounding, and document retrieval—where both the input and the user instruction determine the retrieval target.

In this context, leveraging multimodal large language models (MLLMs) as encoders for UME is a promising new approach (Jiang et al., 2024; Gu et al., 2025; Lan et al., 2025). MLLMs can generate instruction-aware representations that incorporate long context and nuanced user guidance (Peng

---

[*]Equal contribution.
[†]Work done during the internship at Meta. Email: xuanming.cui@ucf.edu

et al., 2024). However, while state-of-the-art models on these benchmarks have advanced through techniques such as hard negative sampling (Lee et al., 2025; Thirukovalluru et al., 2025; Lan et al., 2025; Lin et al., 2025), additional training stages (Gu et al., 2025; Yu et al., 2025; Lin et al., 2025), external data (Yu et al., 2025; Chen et al., 2025b), and improved embedding extraction (Faysse et al., 2024), a central challenge remains: understanding instructions of varying complexity that require different levels of reasoning. Addressing this challenge calls for leveraging the generative capacity of MLLMs, rather than restricting them to encoder-only models.

Our approach is motivated by a closer examination of the tasks in MMEB-v2 (Meng et al., 2025), revealing that many, such as VQA, classification, visual grounding, and composed retrieval, require substantial reasoning capabilities. For example, the first case in Fig. 1 (from the MMEB-v2 RefCOCO dataset) involves retrieving the image region referred to by the query *"vehicle second closest to camera"*. Rather than simply encoding the query and matching it to the target image, we argue that it is beneficial for the model to first *reason* about the query—identifying, for instance, that it refers to the vehicle with *"bright yellow on the upper half and deep blue on the lower"*. For such complex tasks, we believe that explic-

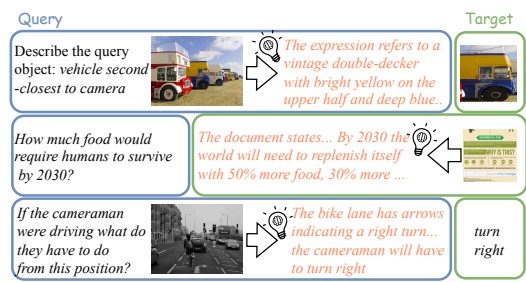

Figure 1: Given a multi-modal input, we want to first *think* about the desired embedding content. Then, the representation is obtained by conditioning on both original input and the thinking result.

itly augmenting multi-step reasoning and contextual grounding enables the model to achieve more fine-grained and accurate retrieval.

To address these challenges, we propose a *Think-Then-Embed* (TTE) framework for the UME task. Our key idea is to introduce an explicit *thinking* stage before embedding, in which the model generates reasoning traces based on the given instruction. Prior work has shown that intermediate reasoning processes, such as Chain-of-Thought (CoT) (Wei et al., 2023b), can significantly improve the accuracy of language model generation. We hypothesize that a similar benefit can be achieved for multimodal representation learning, since the embeddings produced by MLLMs are inherently conditioned on the sequence of preceding tokens. In this work, we investigate how explicit chain-of-thought reasoning can enhance universal multimodal embeddings, by enabling the model to better interpret, follow complex task instructions, and to reduce retrieval asymmetry.

In this paper, we made several contributions. First, we proposed a *Think-Then-Embed* (TTE) framework that by exploiting reasoning traces generated by a powerful reasoner (*e.g.* Qwen2.5 VL 72B), it can achieve state-of-the-art performance on MMEB V2 (Meng et al., 2025) with a smaller embedder (Qwen2 7B), surpassing close-sourced models trained with additional data. This provides an effective way for retrieval test-time-scaling, by flexibly incorporating strong external reasoner.

Second, we study how a backbone model can be effectively used both as a reasoner and as an embedder. We experimented with two approaches. First, we use the backbone as a zero-shot reasoner while fine-tuning a copy of the same backbone as the embedder. This method yields noticeable improvements on smaller datasets, but the gains diminish as the dataset size increases. Motivated by this limitation, our second approach involves fine-tuning a second copy of the backbone (same size as the embedder, either 2B or 7B) as a strong reasoner, using reasoning traces generated by the 72B reasoner model as the training data. The resulting SFT-ed reasoner performs effectively across the entire MMEB v2 benchmark, achieving the best performance among all open source models.

Finally, to improve efficiency of both inference and number of parameters in the TTE framework, we explore unifying reasoning and embedding creation within a single unified model. Although prior work (Yu et al., 2025) also investigated the unification of generation and contrastive learning, it treats generation and embedding as separate, unrelated tasks. In contrast, our focus is on enhancing embedding through reasoning, that is, generate first, then embed. We explored two strategies: (1) joint contrastive-autoregressive training of the same backbone for reasoning and embedding tasks, and (2) a two-stage autoregressive-then-contrastive training with a dedicated embedding head on top of the reasoner. Empirically, we find that the two-stage approach consistently outperforms joint training, almost halving the overall parameters, while not degrading end-to-end TTE retrieval performance.

## 2 RELATED WORK

**Multimodal Embeddings.** The multimodal representation paradigm has been popularized by large-scale foundational models such as CLIP (Radford et al., 2021), MetaCLIP (Xu et al., 2024), BLIP (Li et al., 2023a), and SigLIP (Zhai et al., 2023), which encode images and texts using separate uni-modality encoders and enforce alignment using contrastive objectives.

**Universal Multimodal Embedding.** Recently there has been growing interest in developing Universal Multimodal Embedding (UME) (Gu et al., 2025; Jiang et al., 2024; Meng et al., 2025), where the embedding depends both on the query and task instruction. Representative examples include VLM2Vec (Jiang et al., 2024), which proposes the Massive Multimodal Embedding Benchmark (MMEB), comprising a wide range of cross / multimodal retrieval tasks, as well as non-conventional retrieval tasks such as VQA, grounding, and classification. Subsequently, MMEB-V2 (Meng et al., 2025) is proposed, extending MMEB-V1 to include video and visual document (visdoc) tasks.

**MLLM-based Embedding Models.** Recent studies have gone beyond dual encoder setups, building embedding models directly on top of powerful Multimodal Language Models (MLLMs) (Jiang et al., 2024; Meng et al., 2025; Gu et al., 2025; Yu et al., 2025; Thirukovalluru et al., 2025). Existing work on MLLM-based embedding typically explores a training strategy such as text-continual contrastive training (Gu et al., 2025; Yu et al., 2025; Chen et al., 2025a), hard negative mining approach (Lee et al., 2025; Lan et al., 2025; Thirukovalluru et al., 2025), additional training data (Zhou et al., 2024; Chen et al., 2025b), and architectural design (e.g. with / without causal mask) (Lee et al., 2025; Chen et al., 2025a). However, these methods all treat MLLMs solely as encoders, while overlooking their generative capability obtained from pre-training. In contrast, we explore an orthogonal approach, where we leverage MLLMs for both generative reasoning and representation learning.

**LLM-based Query Rewriting.** Frequently used in text-based retrievals, query rewriting (Ma et al., 2023) is an effective approach to improve retrieval accuracy. Popular trends include prompt-based rewriting (Wilson et al., 2025; Ye et al., 2023), iterative or Reinforcement Learning (RL)-guided (Cao et al., 2025; Zhu et al., 2025) query rewritings, and multi-reformulation approaches (Kostric & Balog, 2024; Dhole et al., 2024). However, these works all focus on text-based retrievals using external retrievers, where no learned embedding is involved.

Recently, Bai et al. (2025b) applied the idea of query rewriting to uni-modal encoder-based models (e.g. CLIP) for text-to-video retrieval, by enriching textual queries with additional context to bridge information asymmetry. In contrast, our work focuses on leveraging reasoning to improve the quality of general instruction-following multimodal embeddings, which can apply to both query and target.

## 3 PRELIMINARY: MLLM-BASED UNIVERSAL MULTIMODAL EMBEDDING

We briefly introduce MLLM-based Universal Multimodal Embedding (UME). In UME, every query $q$ and target $t$ is a triplet of an optional visual input, a textual input, and a pre-defined task instruction, written as $\langle \mathcal{V}, \mathcal{T}, [\texttt{Ins}] \rangle$. Both query and target are passed into the same MLLM separately to obtain the corresponding embedding, defined by the equation below:

$$\mathbf{h}_q = \texttt{Pooling}(f_\theta(\mathcal{V}_q, [\texttt{Ins}_q], \mathcal{T}_q)), \quad \mathbf{h}_t = \texttt{Pooling}(f_\theta(\mathcal{V}_t, [\texttt{Ins}_t], \mathcal{T}_t)),$$

where $f_\theta$ denotes the MLLM embedder and $\texttt{Pooling}$ refers to the pooling operation for aggregating MLLM's hidden states into the final embedding. Following previous works (Jiang et al., 2024; Gu et al., 2025), we use the last token's hidden state for final embedding. During training, we adopt the standard uni-directional $q \rightarrow t$ InfoNCE loss as follow:

$$\mathcal{L}_{\text{InfoNCE}} = -\frac{1}{N} \sum_{i=1}^{N} \log \frac{\phi(\mathbf{h}_q^i, \mathbf{h}_t^i)}{\sum_{j=1}^{N} \phi(\mathbf{h}_q^i, \mathbf{h}_t^j)}, \quad \phi(\mathbf{h}_q, \mathbf{h}_t) = \exp(\frac{1}{\tau} \cos(\mathbf{h}_q, \mathbf{h}_t)),$$

where $\cos$ denotes the cosine similarity function, and $\tau$ is the temperature hyper-parameter.

## 4 THINK-THEN-EMBED

We propose our *Think-then-Embed* (TTE) framework for enhancing Universal Multimodal Embeddings (UME) by incorporating an explicit reasoning step prior to the encoding step. TTE consists

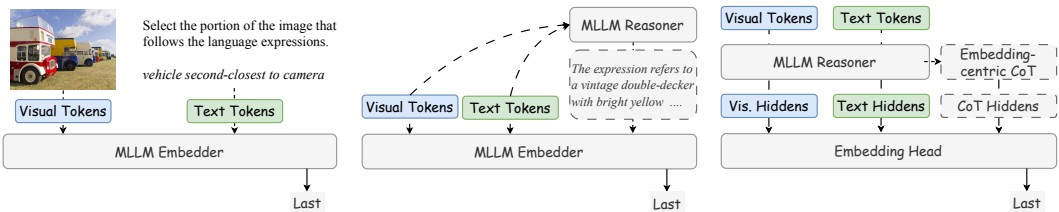

(a) Existing MLLM-based embedding approach. The MLLM embedder directly encode visual and textual queries, producing the embedding from the last token.

(b) TTE with a separate MLLM reasoner. The MLLM embedder takes in both the original inputs and the embedding-centric CoT generated by the MLLM reasoner.

(c) TTE with unified reasoning and embedding. After the MLLM reasoner finishes generation, its entire hidden states are passed to the pre-trained embedding head.

Figure 2: Pipeline comparison between existing MLLM-based embedding (a) and proposed approach (b, c).

of a reasoner, that generates an Embedding-Centric Reasoning (ECR) trace, and an embedder, that produces task-specific representations conditioned on both the original input and the reasoning trace generated by the reasoner. Fig. 2 compares the standard approach of directly encoding multimodal inputs for embedding (2a) with our proposed approach TTE (2b, 2c).

Formally, given a multimodal input $\langle \mathcal{V}, \mathcal{T}, [\texttt{Ins}] \rangle$, we obtain its embedding $\mathbf{h}$ by:

$$\mathbf{h} = \texttt{Pooling}(f_\theta(\mathcal{V}, [\texttt{Ins}], \mathcal{T}, \psi)), \quad \psi = g_\omega(\mathcal{V}, [\texttt{Ins}], \mathcal{T}),$$

where $g_\omega$ and $\psi$ are the reasoner MLLM and its embedding-centric reasoning trace, respectively.

## 4.1 EMBEDDING-CENTRIC REASONING

Here, we introduce the concept of *Embedding-Centric Reasoning (ECR)*, a form of intermediate reasoning tailored to improve embedding quality. Inspired by *chain-of-thought (CoT)* reasoning (Wei et al., 2023b) in complex problem solving, ECR are generative reasoning traces that explicitly support the production of target embeddings. For example, for VQA query, the ECR represents the model's step-by-step reasoning to understand the query, whereas for grounding query, it captures a detailed description of the referred object along with its surrounding visual context.

Formally, for a batch of $(q, t)$ pair, $\mathbf{h}_q$ and $\mathbf{h}_t$ are conditioned on $\psi$, the ECR $\psi$ can be described as

$$\psi_i^\star \in \arg\max_\psi \ \log \frac{\phi(\mathbf{h}_q^i, \mathbf{h}_t^i)}{\sum_j \phi(\mathbf{h}_q^i, \mathbf{h}_t^j)}.$$

In this work, we simplify the learning of ECR, by using manually designed task prompt and format to prompt reasoner MLLM to generate ECR. We formulate ECR in the form of `<think>···</think>` Final Reasoning, where the reasoner model first outputs some intermediate CoT, then generates the final reasoning. The content for CoT and reasoning can vary in terms of task types. For instance, for QA-based query, the CoT is the standard reasoning process; for simple embedding tasks such as visual document embedding, the CoT is a detailed description of the visual input, and the reasoning is as the summary. While it is also possible to directly optimize reasoner MLLM for ECR generation using retrieval signals (*e.g.* RL-based finetuning), we will leave these explorations for future work. By conditioning the embedder on these task-aligned reasoning traces, we enable the model to construct more semantically aligned and task-aware embeddings.

**Inference cost.** We note that, CoT is known to have higher inference cost. In the following subsections, we explore different approaches to construct the reasoner $g_\omega$ that can lower the cost but still improve the performance significantly.

## 4.2 TTE WITH TEACHER REASONER

For our TTE framework, we employ a powerful MLLM (e.g., Qwen2.5VL-72B) as the teacher reasoner, while keeping the embedder model lightweight (Qwen2VL-2B or 7B). We refer to this setup as $\text{TTE}_t$. We utilize the ECR reasoning traces generated by the reasoner for both training and evaluation of the embedder. Although this setup may seem computationally expensive, many real-world

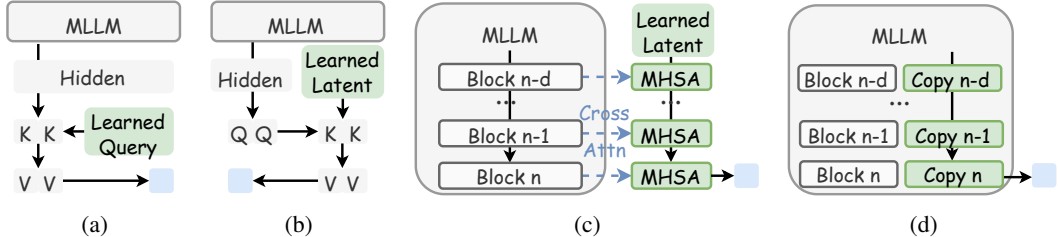

Figure 3: Embedding head designs: (a) Attention Pooling with learnable query. (b) NV-Embed-style (Lee et al., 2025) pooler. (c) Qformer-style embedding head, and (d) Embedding head with self attention blocks initialized from the backbone MLLM. Green denotes trainable components. Q, K and V denote query, key, and value in attention mechanism. MHSA refers to MultiHead Self Attention, and ■ refers to output embedding.

retrieval tasks only require a one-time offline inference step to generate ECR (e.g., detailed descriptions of visual documents) when constructing the retrieval index. Importantly, the reasoner is not involved in online retrieval for existing data points, and only needs to be run once for new data points. We empirically show that $TTE_t$ achieves superior performance on MMEB-v2 without requiring additional training data, advanced training techniques or changes to the model architecture.

### 4.3 TTE WITH FINETUNED REASONER

While our TTE framework supports seamless integration with large MLLMs as the teacher reasoner, we are also interested in exploring whether comparable performance gains can be achieved with a smaller reasoner − specifically, by using the same MLLM backbone as the embedder. As an initial study (Fig. 4), we employ the backbone model itself to generate zero-shot ECRs, which are then used to train the embedder for retrieval. This setup results in noticeable performance gains across classification, retrieval, VQA, and grounding tasks.

Encouraged by these results, we further finetune a dedicated, student reasoner, initializing it from the same backbone as the embedder (2B or 7B). We refer this setting as $TTE_s$. Specifically, we finetune the small reasoner using ECR traces generated by the teacher reasoner (e.g., Qwen2.5VL-72B). Given a multimodal input $\langle \mathcal{V}, [\texttt{Ins}], \mathcal{T} \rangle$, we maximize the conditional likelihood of the ECR $\psi$ by optimizing the standard negative log-likelihood (NLL) loss in standard LM finetuning:

$$\mathcal{L}_{\text{SFT}}(\omega) = -\frac{1}{T} \sum_{t=1}^{T} \log p_\omega(\psi_t \mid \mathcal{V}, [\texttt{Ins}], \mathcal{T}, \psi_{t'<t}).$$

### 4.4 TTE WITH UNIFIED REASONER AND EMBEDDER

The TTE framework described so far rely on separately trained reasoners and embedders, where the reasoner's ECR tokens are encoded together with the query to form embeddings. To reduce the computational overhead introduced by the additional reasoner, we explore unifying reasoning and embedding generation within a shared backbone, enabling embeddings to be produced in a single forward pass. We consider two approaches: (1) joint SFT-contrastive training of a shared reasoner–embedder backbone, and (2) a reasoner augmented with an embedding head. We begin with (1), training reasoning and embedding jointly in a multi-task learning setup. However, this configuration consistently resulted in performance degradation, due to the challenges in training curriculum. The implementation details and results for this approach are provided in Appendix B. For the rest of the section, we focus on design (2), as it offers greater flexibility in training control.

**Reasoner with Embedding Heads.** Our key design principle is that hidden states generated during reasoning can also be reused for downstream representation extraction. However, the two tasks should have specialized parameters so that the embedding training objective will not interfere with reasoning, and vice versa. Therefore, we adopt a two-stage training strategy for the unified model. In the first stage, we fully fine-tune the backbone as a reasoner on the curated ECR dataset. In the second stage, we freeze the backbone and train only the embedding head on top of it. This design enables the unified model to produce embeddings in a single forward pass, as illustrated in Figure 2c.

We refer this setting is $\text{TTE}_u$. As we decouple the contrastive and generative objective, there is no contrastive gradient flowing into the reasoning MLLM, and vice versa.

**A Systematic Study of Pluggable Embedding Head.** We present the first systematic study of embedding head designs that can be seamlessly plugged into a frozen backbone MLLM. This stands in contrast to the focus of prior LLM-based embedding models (Lee et al., 2025; Lin et al., 2025; Jiang et al., 2024), which primarily rely on either full finetuning or parameter-efficient approaches based on LoRA (Hu et al., 2021). As illustrated in Fig. 3, we evaluate four embedding head designs: (1) Learnable attention queries over backbone hidden states (Fig. 3a); (2) Learnable latent context with backbone hidden states as queries; (3) A QFormer-style (Li et al., 2023a) embedding head patched onto the last $n$ layers of the backbone (Fig. 3c). (4) Repeat the last $n$ layers of the backbone to formulate N new layers as the embedding head (Fig. 3d).

We present a detailed experimental comparison of the four designs in Section 7.3.

## 5 EXPERIMENTS

### 5.1 EXPERIMENTAL SETUP

**Datasets.** We conduct our experiments on both MMEB-V2 (Meng et al., 2025) and MMEB-V1 (Jiang et al., 2024) datasets. MMEB-V1 comprises 20 in-distribution (IND) tasks and 16 out-of-distribution (OOD) tasks, spanning 4 meta-tasks: classification, VQA, retrieval, and grounding. MMEB-V2 extends MMEB-V1 by introducing additional video and visual document (visdoc) retrieval tasks. Video tasks including video QA, classification, retrieval, and video moment retrieval. Visdoc involves supportive document retrieval, where the task is to retrieve supporting documents given the query. Overall, MMEB-V2 includes a total of 78 test tasks. We report NDCG@5 for visdoc retrieval and Precision@1 for image and video tasks (Meng et al., 2025).

**Models.** We employ Qwen2VL 2B and 7B as the backbone, and introduce three variants of TTE: TTE with a large teacher reasoner ($\text{TTE}_t$), TTE with a small, SFT-ed student reasoner ($\text{TTE}_s$), and TTE with a unified reasoner and embedder ($\text{TTE}_u$). We compare our approach against several baselines, including dual-encoder methods such as CLIP (Radford et al., 2021), SigLIP (Zhai et al., 2023), UniIR (Wei et al., 2023a), and MagicLens (Zhang et al., 2024). We also directly compare with VLM2Vec-V1 (Jiang et al., 2024) and VLM2Vec-V2 (Meng et al., 2025) with Qwen2VL as the backbone MLLM. Additionally, we evaluate against recent MLLM-based embedding approaches, including UniME (Gu et al., 2025), LLaVE (Lan et al., 2025), and B3 (Thirukovalluru et al., 2025).

**Training Details.** For both MMEB-V1 and MMEB-V2, we use a global batch size of $8192$, with a learning rate of $2e^{-4}$ for the backbone MLLM and $5e^{-4}$ for the embedding head. The temperature for contrastive loss is set to $0.02$. We train the backbone using LoRA (Hu et al., 2021) with rank 16 and alpha 64. Following VLM2Vec, we employ GradCache (Gao et al., 2021) to increase the per-device batch size. We train for 1 epoch on MMEB-V1, and 2.3 epochs for MMEB-V2. For MMEB-V2, we follow the same data weighing setup as in VLM2Vec-V2, and adopt its interleaved sampling strategy, where one global batch is splitted into $n$ sub batches, each from one dataset.

For supervised finetuning of the ECR Reasoner, we perform full parameter finetuning of the MLLM while keeping the visual encoder frozen, using a learning rate of $2e^{-5}$, a global batch size of $128$, and training for an epoch. We use DeepSpeed (Aminabadi et al., 2022) ZeRO 1 for optimizer offloading.

## 6 RESULTS

### 6.1 MAIN RESULTS

**Results on MMEB-V1.** Table 1 compares the performance of our models with recent approaches on MMEB-V1. The encoder-based baselines are evaluated in a zero-shot setting, while the remaining models are trained on the MMEB-V1 training data. Our proposed approach TTE achieves the best performance in both the 2B and 7B categories, with $\text{TTE}_t$ outperforming the next best approach by $6\%$. Compared to VLM2Vec-V1, $\text{TTE}_s$, $\text{TTE}_u$ and $\text{TTE}_t$ achieve substantial improvements of $7.5\%$, $7.4\%$ and $12.7\%$, respectively, on the 7B embedding model.

| Model | Backbone | Per Meta-Task Score | | | | Average Score | | |
|---|---|---|---|---|---|---|---|---|
| | | Classification | VQA | Retrieval | Grounding | IND | OOD | Overall |
| # of datasets → | | 10 | 10 | 12 | 4 | 20 | 16 | 36 |
| *Encoder-Based Baselines* | | | | | | | | |
| CLIP (Radford et al., 2021) | - | 42.8 | 9.1 | 53.0 | 51.8 | 37.1 | 38.7 | 37.8 |
| SigLIP (Zhai et al., 2023) | - | 40.3 | 8.4 | 31.6 | 59.5 | 32.3 | 38.0 | 34.8 |
| UniIR (CLIP$_{SF}$) (Wei et al., 2023a) | - | 44.3 | 16.2 | 61.8 | 65.3 | 47.1 | 41.7 | 44.7 |
| MagicLens (Zhang et al., 2024) | - | 38.8 | 8.3 | 35.4 | 26.0 | 31.0 | 23.7 | 27.8 |
| *~ 2B Model Size* | | | | | | | | |
| VLM2Vec-V1 (Jiang et al., 2024) | Qwen2VL | 59.0 | 49.4 | 65.4 | 73.4 | 66.0 | 52.6 | 59.3 |
| UniME (Gu et al., 2025) | LLaVA-1.6 | 54.8 | 55.9 | 64.5 | 81.8 | 68.2 | 52.7 | 64.2 |
| LLaVE (Lan et al., 2025) | Aquila-VL | 62.1 | 60.2 | 65.2 | 84.9 | 69.4 | 59.8 | 65.2 |
| B3 (Thirukovalluru et al., 2025) | Qwen2VL | 67.0 | 61.2 | 70.9 | 79.9 | 72.1 | 63.1 | 68.1 |
| (Ours) TTE$_s$ | Qwen2VL | 67.6 | 62.7 | 70.7 | 80.0 | 71.4 | 65.2 | 68.7 |
| (Ours) TTE$_u$ | Qwen2VL | 69.7 | 60.8 | 71.4 | 78.4 | 71.6 | 65.1 | 68.8 |
| (Ours) TTE$_t$ | Qwen2VL | **72.6** | **74.3** | **72.6** | **85.2** | **80.5** | **67.0** | **74.5** |
| *~ 7B Model Size* | | | | | | | | |
| VLM2Vec-V1 (Jiang et al., 2024) | Qwen2VL | 62.6 | 57.8 | 69.9 | 81.7 | 65.2 | 56.3 | 65.8 |
| UniME (Gu et al., 2025) | LLaVA-OV | 66.8 | 66.6 | 70.5 | 90.9 | 74.6 | 65.8 | 70.7 |
| LLaVE (Lan et al., 2025) | LLaVA-OV | 65.7 | 65.4 | 70.9 | **91.9** | 75.0 | 64.4 | 70.3 |
| B3 (Thirukovalluru et al., 2025) | Qwen2VL | 70.0 | 66.5 | 74.1 | 84.6 | 75.9 | 67.1 | 72.0 |
| QQMM (Xue et al., 2025) | LLaVA-OV | 69.9 | 70.0 | 72.1 | 86.0 | 77.2 | 66.6 | 72.5 |
| (Ours) TTE$_s$ | Qwen2VL | 70.5 | 71.5 | 73.5 | 83.9 | 76.8 | 68.9 | 73.3 |
| (Ours) TTE$_u$ | Qwen2VL | 71.6 | 72.0 | 72.5 | 82.1 | 76.7 | 68.7 | 73.2 |
| (Ours) TTE$_t$ | Qwen2VL | **77.4** | **78.6** | **75.9** | 89.0 | **82.7** | **73.3** | **78.5** |

Table 1: Results on the MMEB-V1 benchmark (Jiang et al., 2024). The scores are averaged per meta-task. Best/2nd-best performance across both 2B and 7B categories are in **bold**/underlined.

| Model | Image | | | | | Video | | | | | VisDoc | | | | | All |
|---|---|---|---|---|---|---|---|---|---|---|---|---|---|---|---|---|
| | CLS | QA | RET | GD | Overall | CLS | QA | RET | MRET | Overall | VDRv1 | VDRv2 | VR | OOD | Overall | |
| # of Datasets → | 10 | 10 | 12 | 4 | 36 | 5 | 5 | 5 | 3 | 18 | 10 | 4 | 6 | 4 | 24 | 78 |
| **Close-sourced Models or w/ Additional Data** | | | | | | | | | | | | | | | | |
| seed-1.6-embedding | 76.1 | 74.0 | 77.9 | 91.3 | 77.8 | 55.0 | 60.9 | 51.3 | 53.5 | 55.3 | **89.5** | 60.8 | 87.9 | 44.4 | **76.8** | 71.3 |
| RzenEmbed-v1-2B | 65.3 | 61.7 | 73.8 | 77.9 | 68.5 | 45.6 | 47.5 | 38.3 | 36.7 | 42.6 | 87.0 | 57.6 | 85.4 | 43.3 | 74.4 | 64.4 |
| Ops-MM-embedding-v1-2B | 68.1 | 65.1 | 69.2 | 80.9 | 69.0 | 53.6 | 55.7 | 41.8 | 33.7 | 47.6 | 87.0 | 57.6 | 85.4 | 43.3 | 74.4 | 63.4 |
| GME-2B | 54.4 | 29.9 | 66.9 | 55.5 | 51.9 | 34.9 | 42.0 | 25.6 | 32.4 | 33.9 | 86.1 | 54.0 | 82.5 | 43.1 | 72.7 | 54.1 |
| ColPali v1.3-3B | 40.3 | 11.5 | 48.1 | 40.3 | 34.9 | 26.7 | 37.8 | 21.6 | 25.5 | 28.2 | 83.6 | 52.0 | 81.1 | 43.1 | 71.0 | 44.4 |
| Ops-MM-embedding-v1-7B | 69.7 | 69.6 | 73.1 | 87.2 | 72.7 | 59.7 | 62.2 | 45.7 | 43.2 | 53.8 | 80.1 | 59.6 | 79.3 | 43.3 | 70.3 | 67.6 |
| RzenEmbed-v1-7B | 69.8 | 68.7 | 76.8 | 85.7 | 73.6 | 52.8 | 56.2 | 41.9 | 41.8 | 48.9 | **89.5** | 60.8 | 87.9 | 44.4 | 76.8 | 68.9 |
| GME-7B | 57.7 | 34.7 | 71.2 | 59.3 | 56.0 | 37.4 | 50.4 | 28.4 | 38.2 | 38.6 | 89.4 | 55.6 | 85.0 | 44.4 | 75.2 | 57.8 |
| LamRA-Qwen2-7B | 59.2 | 26.5 | 70.0 | 62.7 | 54.1 | 39.3 | 42.6 | 24.3 | 34.6 | 35.2 | 22.0 | 11.5 | 37.4 | 21.0 | 23.9 | 40.4 |
| LamRA-Qwen2.5-7B | 51.7 | 34.1 | 66.9 | 56.7 | 52.4 | 32.9 | 42.6 | 23.2 | 37.6 | 33.7 | 56.3 | 33.3 | 58.2 | 40.1 | 50.2 | 47.4 |
| **Models Trained on MMEB V2** | | | | | | | | | | | | | | | | |
| VLM2Vec-V2-2B | 62.9 | 56.3 | 69.5 | 77.3 | 64.9 | 39.3 | 34.3 | 28.8 | 38.5 | 34.9 | 75.5 | 44.9 | 79.4 | 39.4 | 65.4 | 58.0 |
| (Ours) TTE$_s$ (2B) | 67.9 | 66.6 | 70.2 | 84.1 | 70.1 | 47.3 | 49.1 | 34.4 | 33.2 | 42.1 | 77.5 | 53.2 | 83.2 | 41.1 | 68.8 | 63.1 |
| (Ours) TTE$_u$-2B | 68.5 | 65.7 | 70.9 | 83.1 | 70.1 | 47.6 | 49.2 | 33.3 | 33.5 | 41.7 | 78.0 | 54.0 | 84.5 | 61.1 | 73.9 | 63.3 |
| (Ours) TTE$_t$-2B | 76.6 | 76.8 | 71.5 | 87.2 | 76.1 | 56.1 | 65.3 | 34.1 | 33.8 | 47.9 | 81.1 | 62.4 | 84.7 | 43.2 | 72.6 | 68.6 |
| VLM2Vec-V2-7B | 65.7 | 61.5 | 70.0 | 85.2 | 68.1 | 45.9 | 33.9 | 27.6 | 39.3 | 36.4 | 78.8 | 52.6 | 82.7 | 42.1 | 69.3 | 61.2 |
| (Ours) TTE$_s$-7B | 69.7 | 72.4 | 74.0 | 90.6 | 74.2 | 49.1 | 56.4 | 37.2 | 46.8 | 46.8 | 84.1 | 62.7 | **91.9** | 47.6 | 76.4 | 68.6 |
| (Ours) TTE$_u$-7B | 69.6 | 72.3 | 73.2 | 90.7 | 73.9 | 51.2 | 59.5 | 36.0 | 38.2 | 47.1 | 83.5 | 63.8 | 91.5 | 66.8 | 80.6 | 68.2 |
| (Ours) TTE$_t$-7B | **76.7** | **78.6** | 74.3 | 89.3 | **77.8** | 57.5 | **68.2** | 38.0 | 39.3 | 52.0 | 83.7 | **63.6** | 91.4 | 50.6 | 76.8 | **71.5** |

Table 2: Results on the MMEB-V2 benchmark (Meng et al., 2025). Best/2nd-best performance across *all* models are in **bold**/underlined. Task abbreviations: CLS (classification), QA (question answering), RET (retrieval), GD (grounding), MRET (moment retrieval), VDR (ViDoRe), VR (VisRAG), and OOD (out-of-domain).

**Results on MMEB-V2.** Table 2 summarizes overall results on MMEB-V2 dataset across the entire leaderboard, at the time of submission. TTE$_t$-7B achieves highest overall score of 71.5% on the MMEB-V2 leaderboard, surpassing recent SOTA models trained with massive external data (*e.g.*, seed-1.6-embedding). Without relying on the teacher reasoner, TTE$_s$ achieves best performance across open-sourced models, for both 2B and 7B variants. Compared to baseline VLM2Vec-V2, TTE shows substantial gain. For instance, TTE$_s$ and TTE$_t$ improve 2B baseline by 5.1% and 10.6%.

Taking a closer look at the performance across modality and tasks, we can observe notable improvement against VLM2Vec-V2 baseline from VQA and classification-based tasks. For instance, TTE$_t$-7B improves video QA performance significantly against VLM2Vec-V2. As for retrieval, we observe larger improvements on video-based retrieval (+5%) as compared to image-based retrieval (*e.g.* +2% on TTE$_t$-2B), possibly due to video-based retrieval is more challenging than image-based, and therefore signifying the role of teacher-generated ECR.

| w/ SFT-ed Reasoner | Per Meta-Task Score | | | | Average Score | | |
|---|---|---|---|---|---|---|---|
| | CLS | VQA | Ret | GD | IND | OOD | All |
| 2B Model | | | | | | | |
| Baseline | 59.0 | 49.4 | 65.4 | 73.4 | 66.0 | 52.6 | 59.3 |
| ✗ | 62.0 | 57.3 | 62.7 | 73.9 | 65.1 | 58.7 | 62.2 |
| ✓ | **67.6** | **62.7** | **70.7** | **80.0** | **71.4** | **65.2** | **68.7** |
| 7B Model | | | | | | | |
| Baseline | 62.6 | 57.8 | 69.9 | 81.7 | 65.2 | 56.3 | 65.8 |
| ✗ | 65.5 | 66.9 | 68.4 | 78.2 | 71.7 | 64.0 | 68.3 |
| ✓ | **70.5** | **71.5** | **73.5** | **83.9** | **76.8** | **68.9** | **73.3** |

Table 3: Ablation on the effect of finetuning $TTE_s$ under MMEB-V1. **w/ SFT-ed Reasoner** denotes whether to apply SFT/zero-shot generation of ECR.

| w/ CoT | Per Meta-Task Score | | | | Average Score | | |
|---|---|---|---|---|---|---|---|
| | CLS | VQA | Ret | GD | IND | OOD | All |
| 2B Model | | | | | | | |
| Baseline | 59.0 | 49.4 | 65.4 | 73.4 | 66.0 | 52.6 | 59.3 |
| ✗ | 71.5 | 73.6 | 72.6 | 83.6 | 79.9 | 66.1 | 73.8 |
| ✓ | **72.6** | **73.9** | **73.1** | **85.2** | **80.5** | **67.0** | **74.5** |
| 7B Model | | | | | | | |
| Baseline | 62.6 | 57.8 | 69.9 | 81.7 | 65.2 | 56.3 | 65.8 |
| ✗ | 75.8 | 78.2 | 72.9 | 85.8 | 80.6 | 71.6 | 76.6 |
| ✓ | **77.5** | **78.6** | **74.7** | **88.9** | **82.1** | **73.2** | **78.1** |

Table 4: Effect of CoT traces in ECR with $TTE_t$. We ablate the same set of ECR but w/ CoT or w/o CoT (only final reasoning).

# 7 ABLATIONS

## 7.1 UNDERSTANDING THE ROLE OF TTE REASONER AND ECR

**Effect of zero-shot `TTE` reasoner.** We begin by analyzing the impact of the model's inherent reasoning capability to the retrieval performance. We first prompt the MLLM reasoner (Qwen2VL-2B) to generate ECR in a zero-shot setting. The generated ECR is then combined with the raw query for embedder training using the same Qwen2VL-2B backbone. We conduct the ablation on a subset of datasets within MMEB V1, where constrastive training is performed individually for each dataset. Fig. 4 shows the corresponding results. We can observe that incorporating self-generated zero-shot ECR leads to notable performance improvements across all tasks.

**Effect of finetuning `TTE` reasoner.** While we find that the zero-shot reasoner can generally improve embedding performance, its effectiveness can still be bottlenecked by the quality of the zero-shot ECR. In Table 3, we present ablation results on finetuning the TTE reasoner on MMEB-V1, where we observe a significant improvement after finetuning, with an absolute gain of over 6%.

**Effect of intermediate CoT in ECR.** To examine the role of reasoning within ECR, we conduct an ablation by including or excluding intermediate CoT during training and testing, with results shown in Table 4. Overall, incorporating CoT leads to consistent performance gains. We observe clear improvements on classification, retrieval, and grounding, with only marginal gains on VQA—an expected outcome since VQA primarily requires the final answer, whereas retrieval and grounding benefit from richer intermediate context. Notably, the improvements are larger for 7B than 2B models, likely due to the stronger language capacity of larger models.

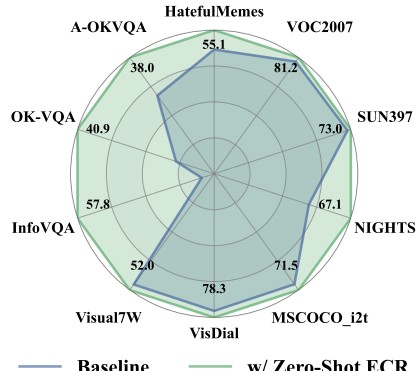

Figure 4: Baseline under Qwen2VL-2B with/without zero-shot ECR on MMEB-V1.

**Analysis on the role of ECR.** We conducted more analysis to understand the role of ECR in Fig. 5.

A natural question is whether the ECR text alone is sufficient for retrieval, potentially eliminating the need to train a dedicated encoder. However, our experiments on MMEB show that this is not the case. Specifically, when we use the same ECR and apply a text-to-text (T2T) retrieval baseline—by directly encoding the ECR with a strong off-the-shelf text encoder, Jina-V3 (Sturua et al., 2024) – the performance is significantly worse.

Another concern is that imperfect ECRs might introduce noise that propagates into the Embedder, degrading retrieval performance. Interestingly, we find that our TTE Embedder is robust to such noise: it learns to extract useful information from the ECR without blindly relying on it. In fact, although ECRs alone do not yield high retrieval precision, they provide valuable signals that our Embedder can leverage. We observe similar trends across all task types. These findings indicate that our Embedder is robust to noisy ECRs, and retrieval performance can benefit even from imperfect reasoning traces. Meanwhile, we also note that the observed robustness does not imply that the embedder is invariant to the quality of ECRs. In fact, the robustness range we measure is within the

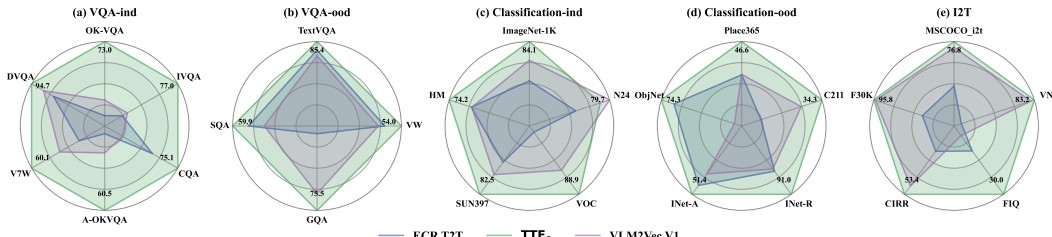

Figure 5: Results on T2T evaluation on generated ECR, versus $TTE_s$ and VLM2Vec-V1 on MMEB V1.

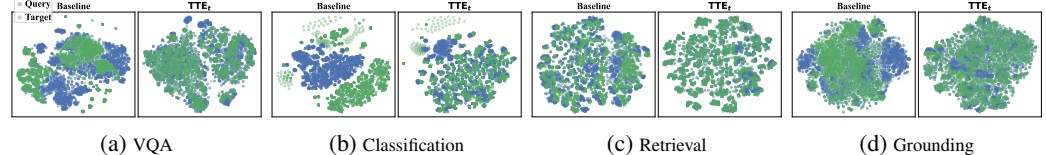

(a) VQA     (b) Classification     (c) Retrieval     (d) Grounding

Figure 6: $t$-SNE visualization (perplexity $= 30$) of query and target embeddings between baseline VLM2Vec 7B (left) and ours ($TTE_t$, right). Zoom in for better visual effect.

*same* reasoner: the embedder outperforms directly using the ECRs themselves (e.g., for VQA-style tasks where retrieval does not rely on dense visual embeddings). However, using a *different* reasoner can still impact the embedder's performance, as shown in Table 3 and between $TTE_s$ and $TTE_t$.

## 7.2 VISUALIZATION

In Fig. 6 we show a side-by-side $t$-SNE visualization comparing embedding produced by the baseline VLM2Vec and $TTE_t$ on MMEB-V1. For each task type, we select 3 distinct subtasks. We clearly observe larger overlap between query and target embedding for $TTE_t$, as compared to VLM2Vec.

## 7.3 EMBEDDING HEAD DESIGN

**Architecture Design.** In Table 5, we investigate the design of embedding heads. We first explore simple (depth=1) trainable pooler, including an attention pooler with learnable queries and an NV-Embed–style latent pooler with learnable context. However, both approaches perform significantly worse than $TTE_s$, and increasing the number of queries (n_queries $= 1 \rightarrow 16$) yields little improvement. We hypothesize that the hidden states from the final layer may not be optimal for capturing latent representations, as these layers are primarily tuned for providing discriminative features for token predictions.

| Model | Per Meta-Task Score | | | | Average Score | | |
|---|---|---|---|---|---|---|---|
| | CLS | VQA | Ret | GD | IND | OOD | Overall |
| $TTE_s$ | 67.6 | **62.7** | 70.7 | **80.0** | 71.4 | 65.2 | 68.7 |
| w/ Simple Attention Pooler | | | | | | | |
| n_queries $= 1$ | 48.0 | 45.4 | 45.8 | 53.1 | 50.5 | 42.9 | 47.1 |
| n_queries $= 8$ | 48.3 | 46.1 | 45.9 | 60.1 | 51.3 | 44.3 | 48.2 |
| n_queries $= 16$ | 49.7 | 47.6 | 47.8 | 56.7 | 53.1 | 44.5 | 49.3 |
| w/ NV-Embed-style Pooler | | | | | | | |
| n_latent_value $= 512$ | 39.3 | 46.3 | 42.7 | 57.6 | 48.2 | 39.7 | 44.4 |
| w/ QFormer-style Embedding Head | | | | | | | |
| n_layers/last_n $= 4/4$ | 65.2 | 57.8 | 68.0 | 79.8 | 68.8 | 61.8 | 65.7 |
| n_layers/last_n $= 8/8$ | 65.8 | 60.7 | 69.8 | 78.7 | 70.0 | 63.6 | 67.1 |
| w/ Self-Initialized MHSA | | | | | | | |
| n_layers/last_n $= 4/4$ | 67.0 | 59.3 | 70.2 | 78.5 | 71.2 | 62.2 | 67.2 |
| n_layers/last_n $= 8/8$ | 69.7 | 60.8 | **71.4** | 78.4 | **71.6** | 65.1 | **68.8** |
| n_layers/last_n $= 8/16$ | **70.8** | 58.1 | 68.8 | 83.4 | 69.0 | **66.8** | 68.0 |

Table 5: Embedding head ablations on MMEB-V1.

To better exploit representations from a frozen MLLM, we shift focus to earlier layers (e.g., the 8th-to-last) and equip them with more expressive multi-layer embedding heads. Specifically, we compare a QFormer-style head with a self-initialized multi-head self-attention (MHSA) module. Empirically, the latter achieves superior performance compared to the QFormer-style head, which requires training from scratch. When self-initialized with the last n_layers $= 8$, the MHSA head reaches performance on par with a separately fine-tuned embedder ($TTE_s$).

**The choice of last–$n$ layer for embedding.** Most prior work adopts the final hidden states of an MLLM as embeddings, but this design choice is rarely scrutinized. We hypothesize that the last layer may not be optimal for multimodal retrieval: while it is specialized for token-level discrimination, intermediate layers may retain richer semantic structure. To validate our hypothesis, we ablate the

choice of bottom-$n$ layers under two settings: (1) *Attention Pooling* with a frozen backbone, and (2) *LoRA+Last Token*, where LoRA is applied and the final token embedding is used.

We report overall performance on MMEB v1 in Table 6. Under attention pooling, performance improves modestly as $n$ increases from 1 to 8, then drops at $n = 16$. This trend supports our hypothesis: when the backbone is frozen, intermediate layers can yield slightly more robust embeddings than the final layer. However, the gains are small, indicating that most of the improvement stems from the capacity of the embedding head itself (i.e., additional trainable parameters), rather than layer selection alone.

| Last $n$ | Attn Pooling | LoRA+Last |
|---|---|---|
| 1 | 47.1 | 68.7 |
| 4 | 47.8 | 68.4 |
| 8 | 49.6 | 67.2 |
| 16 | 46.3 | 59.5 |

Table 6: Effect of using bottom-$n$ transformer layers for embedding on MMEB v1 (Qwen2-VL 2B).

In contrast, the LoRA+Last Token setting exhibits the opposite trend—performance declines as $n$ increases. Since LoRA injects trainable rank updates into each transformer block, selecting a larger bottom-$n$ effectively reduces the number of blocks available for adaptation, limiting the model's flexibility. Consequently, the final layer performs best under LoRA fine-tuning, whereas intermediate layers can be slightly preferable when the backbone is frozen.

## 7.4 IMPACT OF REASONING MLLM

A natural question is whether stronger teacher models (e.g., Gemini2.5 Pro (**?**)) can further improve embedding quality through higher-quality reasoning traces. To study this, we run additional experiments on MMEB v2 using Qwen2-VL (7B) as the backbone embedder, while varying the teacher model that generates reasoning traces at evaluation time. Besides Qwen2.5-VL (72B), we consider Qwen2.5-VL (32B) and Gemini2.5-Pro, a state-of-the-art MLLM for visual understanding. We focus on retrieval tasks, where the impact of teacher quality is more meaningful than in VQA.

| Task | Qwen2.5-VL 32B | Qwen2.5-VL 72B | Gemini2.5 Pro |
|---|---|---|---|
| i-ret | 74.1 | 74.3 | 73.6 |
| i-grounding | 88.1 | 89.3 | 90.6 |
| v-ret | 35.8 | 37.6 | 41.8 |
| v-moment_ret | 37.8 | 39.3 | 53.4 |
| visdoc | 76.3 | 76.8 | 76.1 |

Table 7: Effect of using varying capacity of reasoning models at evaluation time on MMEB v2.

We observe that Qwen2.5-VL (32B) performs comparably to Qwen2.5-VL (72B), suggesting that moderate increases in teacher capacity do not necessarily translate to improved retrieval performance. In contrast, Gemini2.5 Pro achieves substantially higher accuracy on video and video-moment retrieval (up to $+14\%$ absolute), but performs slightly worse than Qwen2.5-VL (72B) on image and visdoc retrieval. These trends can be explained by two factors: (1) Qwen2.5-VL (72B) already achieves strong reasoning quality on image and visdoc tasks, but remains relatively weaker on video understanding, where a stronger teacher such as Gemini2.5 Pro offers greater benefit. (2) There is a distribution shift between training traces (from Qwen2.5-VL) and test-time reasoning (from Gemini2.5 Pro), leading to mild degradation on image and visdoc performance. The video tasks, however, are unaffected because the video subsets in MMEB v2 are largely out-of-distribution.

## 8 CONCLUSION

We propose Think-Then-Embed (TTE), a general framework for universal multimodal embedding that leverages a reasoner to "think" before predicting embeddings with an embedder. We first show that using a multimodal LLM as the reasoner can substantially boost the performance of a smaller embedder, demonstrating that CoT-style reasoning also benefits multimodal representation learning. To improve efficiency, we then distill a compact reasoner from the reasoning traces of the large model. The distilled reasoner and the embedder can be trained from the same backbone, yielding both capacity and efficiency gains. Finally, we improve the integration of reasoning and embedding by introducing a pluggable embedding head on top of the reasoner. This design enables embeddings to be produced in a single forward pass, further improving efficiency while halving model parameters. Extensive experiments on the MMEB-V1 and MMEB-V2 benchmarks show that our approach significantly outperforms a range of baselines and recent methods without requiring additional data, validating the effectiveness and robustness of the proposed TTE framework.

ETHICS STATEMENT

We reviewed the ICLR Code of Ethics carefully and do not observe potential concerns for our work.

REPRODUCIBILITY STATEMENT

We made our best efforts to comprehensively document the implementation details. Training hyper-parameters and model architectures are discussed in Section 5.1 and B. We include the dataset construction details including all the example prompts we used in Section C. For evaluation, as mentioned in Section 5.1, we strictly follow the official setup with the codebase released by the original authors. We reveal the full results of each task on the benchmark, without average, in Section D.

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

| Model | Flickr30k-I2T | | Flickr30k-T2I | | MSCOCO | |
|---|---|---|---|---|---|---|
| | R@1 | R@5 | R@1 | R@5 | I2T | T2I |
| $\sim$ 2B Models | | | | | | |
| CLIP(ViT-BigG/14) | 92.9 | - | 79.6 | - | 67.3 | 51.3 |
| UniME | 88.2 | - | 77.0 | - | 66.8 | 49.8 |
| B3++ | 94.9 | - | 82.8 | - | 73.6 | 59.0 |
| VLM2Vec | 89.1 | 98.7 | 68.8 | 90.4 | 68.6 | 71.5 |
| **(Ours)** $TTE_s$ | 94.8 | 99.5 | 83.6 | 95.2 | 75.5 | 74.8 |
| **(Ours)** $TTE_t$ | **95.2** | **99.5** | **84.2** | **95.5** | **77.7** | **78.0** |
| $\sim$ 7B Models | | | | | | |
| EVA-CLIP | 94.5 | - | 80.3 | - | 70.1 | 52.0 |
| UniME (Llava-Next) | 93.4 | - | 81.9 | - | 70.1 | 53.7 |
| B3 7B | 95.9 | - | 85.5 | - | 77.6 | 62.8 |
| VLM2Vec | 94.6 | 99.5 | 80.3 | 95.0 | 73.5 | 78.2 |
| **(Ours)** $TTE_s$ | 95.8 | 99.6 | 86.3 | 97.6 | 76.8 | 80.7 |
| **(Ours)** $TTE_t$ | 96.5 | 99.6 | 86.8 | 98.3 | 78.5 | 82.0 |

Table 8: Results on I2T/T2I retrieval on MSCOCO and zero-shot Flickr30K dataset. We report R@1 and R@5 for Flickr30k, and R@1 for MSCOCO.

| Model | w/ Noisy CL ECR | Per Meta-Task Score | | | | Average Score | | |
|---|---|---|---|---|---|---|---|---|
| | | CLS | VQA | Ret | GD | IND | OOD | All |
| 2B Model Size | | | | | | | | |
| $TTE_s$ | ✗ | 66.8 | 62.1 | 69.5 | 78.6 | 70.6 | 64.1 | 67.7 |
| $TTE_s$ | ✓ | 67.6 | 62.7 | 70.7 | 80.0 | 71.4 | 65.2 | 68.7 |
| $TTE_t$ | ✗ | 71.5 | 72.1 | 72.3 | 82.4 | 79.4 | 65.3 | 73.1 |
| $TTE_t$ | ✓ | 72.6 | 74.3 | 72.6 | 85.2 | 80.5 | 67.0 | 74.5 |
| 7B Model Size | | | | | | | | |
| $TTE_s$ | ✗ | 69.3 | 70.1 | 72.3 | 82.2 | 75.4 | 67.6 | 71.9 |
| $TTE_s$ | ✓ | 70.5 | 71.5 | 73.5 | 83.9 | 76.8 | 68.9 | 73.3 |
| $TTE_t$ | ✗ | 76.1 | 76.0 | 74.5 | 85.2 | 80.8 | 71.3 | 76.6 |
| $TTE_t$ | ✓ | 77.4 | 78.6 | 75.9 | 89.0 | 82.7 | 73.3 | 78.5 |

Table 9: Ablation on the MMEB V1 with or without noisy ECR data for contrastive training.

## A   ADDITIONAL RESULTS AND ANALYSIS

### A.1   RESULTS ON I2T/T2I RETRIEVAL.

We evaluate zero-shot I2T and T2I retrieval on Flickr30k (Plummer et al., 2016) dataset and show the results for MSCOCO retrieval. The results are provided in Tab. 8. We can see both $TTE_s$ and $TTE_t$ surpass baseline VLM2Vec and the recently proposed B3 (Thirukovalluru et al., 2025) by a large margin. For instance, on MSCOCO T2I, the 7B $TTE_t$ achieves 82.0 Precision@1, surpasses B3 (62.8) by almost 20% absolute.

### A.2   CONSTRUCTING ECR DATASET

We construct two separate high-quality ECR for contrastive training for TTE Embedder, and Supervised Finetuning (SFT) for TTE Reasoner, using powerful MLLMs such as Qwen2.5-72B (Bai et al., 2025a). For each task in MMEB, we manually design prompts that take in both query and the ground-truth target. For contrastive training, we aim to blend in a certain amount of "noise" in the training dataset to improve robustness of TTE Embedder against incorrect ECR generation. To prevent label leakage and overfitting, we explicitly instruct the MLLM to rewrite the answers while performing reasoning. Additionally, we only keep 50% of the ground-truth in the training data. For test set, we adopt similar prompts but without the ground-truth target. The process for SFT ECR dataset is similar, except we do not ask the MLLM to rewrite the ground-truth, as we want the TTE Reasoner to learn to output the exact ground-truth as accurate as possible.

In Tab. 9 we show ablation study on whether to apply noisy ECR dataset construction method (i.e. with rephrasing and 50% non-ground-truth ECR) for contrastive training. We can observe that by

| MMEB v1 | Qwen2-VL | Qwen2.5-VL | InternVL3 |
|---|---|---|---|
| VLM2Vec | 59.3 | 58.7 | 57.4 |
| TTE$_t$ | 74.5 | 74.1 | 73.6 |

Table 10: Ablation on different MLLM backbones on MMEB V1.

| Model | $\lambda$ | Per Meta-Task Score | | | | Average Score | | |
|---|---|---|---|---|---|---|---|---|
| | | CLS | VQA | Ret | GD | IND | OOD | All |
| Baseline | - | 57.6 | 47.5 | 65.0 | 71.6 | 61.5 | 55.1 | 58.9 |
| SFT+CL | 1 | 50.7 | 43.2 | 59.5 | 67.1 | 55.6 | 48.3 | 52.3 |
| SFT+CL | 10 | 51.4 | 45.8 | 59.2 | 68.6 | 55.3 | 50.2 | 54.7 |
| SFT+CL | 100 | 52.4 | 48.3 | 61.5 | 68.8 | 56.6 | 51.9 | 56.2 |

Table 11: Ablation on the MMEB V1 with joint SFT+CL training, using Qwen2 2B. Baseline denotes VLM2Vec V1 with Qwen2 2B.

applying noisy ECR data construction for contrastive training, we can obtain 1-2% performance gain across both TTE$_s$ and TTE$_t$, for both model size.

### A.3 IMPACT ON DIFFERENT MLLM BACKBONES

While we conduct main experiments on Qwen2VL, following previous works (Jiang et al., 2024; Thirukovalluru et al., 2025), we additionally evaluated Qwen2.5-VL and InternVL3 on MMEB v1, which we used internally during ablation and model selection. Table 10 summarizes the results.

Across all backbones, TTE$_t$ consistently provides substantial improvements over the VLM2Vec baseline, demonstrating that our method generalizes well beyond a single family of MLLMs. However, we observe that Qwen2-VL remains the strongest-performing backbone among the tested models, despite having lower reported accuracy on downstream visual understanding and reasoning benchmarks. This trend aligns with findings reported by the original MMEB authors, who similarly noted that substituting Qwen2-VL with Qwen2.5-VL did not yield performance gains.[1]

## B    JOINT CONTRASTIVE-AUTOREGRESSIVE TRAINING

**Model and implementation details** Here we explore unifying text generation with embedding, by jointly optimizing for both contrastive and autoregressive objective:

$$\mathcal{L}_{\text{joint}} = \lambda\mathcal{L}_{\text{InfoNCE}} + \mathcal{L}_{\text{SFT}}$$

where $\lambda$ is a hyper-parameter controlling the weight of contrastive (InfoNCE) loss. During training we use the second last token as the embedding token, since it is the last token generated during inference. We apply a simple MLP block on top of the token to obtain the embedding.

**Training details.** We use the same set of hyper-parameters as used in baseline contrastive training: LoRA with rank and alpha equals to 16 and 64, learning rate of $2e^{-4}$, a global batch size of 8192, and train for one epoch.

We provide results in Tab. 11. We can see incorporating SFT objective leads to performance degradation against baseline. As we increase $\lambda$ (reducing impact of SFT), the performance is partially recovered. We conjecture that there may exist conflicting gradients between contrastive and SFT objectives. It could also be due to the expressiveness of the last layer's hidden states as embedding: the last layer in MLLMs may not be optimal for embedding as they are heavily guided to produce discriminative features for token classification.

---

[1]See GitHub discussion: `https://github.com/TIGER-AI-Lab/VLM2Vec/issues/66`.

Given an image and a question, explain step-by-step how the answer can be derived from the image. Please follow the below rules:

- Keep the reasoning concise and grounded in visual or factual evidence. Keep it succinct, within 1-2 sentences.

- Wrap your reasoning in <think> and </think> tags. Then, on the next line, output the final answer, starting with "Answer:".

- Follow the format in the example below.

---

Example: QUESTION: What is the hairstyle of the blond called?

<think>The blonde woman's hair is tied back into a single bunch, which is characteristic of a ponytail.</think> Answer: pony tail

---

Now given the following image and question:

IMAGE: <image>
QUESTION: {query}
Please follow the same format as the example above, providing your reasoning and final answer.

---

Given an image and a question, explain step-by-step how the answer can be derived from the image. Please follow the below rules:

- Keep the reasoning concise and grounded in visual or factual evidence. Keep it succinct, within 1-2 sentences.

- Wrap your reasoning in <think> and </think> tags. Then, on the next line, output the final answer, starting with "Answer:".

- You are provided with the ground-truth answer for reference. Use it to verify your reasoning but do not mention it explicitly in your explanation.

- Rephrase the final answer so that it preserves the exact meaning of the original but may differ in wording or phrasing. Do not add, remove, or alter factual content.

- Follow the format in the example below.

---

Example: QUESTION: What is the hairstyle of the blond called?\nANSER:

<think>The blonde woman's hair is tied back into a single bunch, which is characteristic of a tied-back hair (ponytail).</think> Answer: tied-back hair, pigtail, pony tail.

---

Now given the following image and question:

IMAGE: <image>
QUESTION: {query}\nANSWER: {answer}
Please follow the same format as the example above, providing your reasoning and final answer.

Figure 7: Prompt template for generating ECR data using teacher MLLMs (*e.g.* InternVL3 78B) for VQA tasks. Left: prompt template without ground-truth. Right: prompt template with ground-truth and rephrasing.

---

Given an image, describe briefly how a COCO-style caption can be formed from the image. Follow the instructions below:

- First provide a detailed description of the image, then describe what a COCO-style caption should contain. (Hint: it should focus on the most salient objects and their arrangement in the image.)
- Wrap your reasoning in <think>...</think> (2–3 sentences).
- Then write the final COCO-style caption on the next line as: Answer: <answer>

---

EXAMPLE

<think>The image shows a cozy bedroom with a wooden bed, striped bedsheets, a lamp on the nightstand with its light turned on, and several large pillows arranged neatly from head to foot along the bed. The COCO-style caption should contain the most salient object and arrangement: the pillows.</think> Answer: Several pillows are lined up down the length of a bed.

---

Now answer for the following image: IMAGE: <image>

---

Given an image, describe briefly how a COCO-style caption can be formed from the image. Follow the instructions below:

- First provide a detailed description of the image, then describe what a COCO-style caption should contain. (Hint: it should focus on the most salient objects and their arrangement in the image.)
- Wrap your reasoning in <think>...</think> (2–3 sentences).
- Then write the final COCO-style caption on the next line as: Answer: <answer>
- You are provided with the ground-truth COCO caption for reference. Use it to verify your reasoning but do not mention it explicitly in your explanation.
- Rephrase the final answer so that it preserves the exact meaning of the original but may differ in wording or phrasing. Do not add, remove, or alter factual content.

---

EXAMPLE

GROUND-TRUTH CAPTION: Several pillows are lined up down the length of a bed.

<think>The image shows a cozy bedroom with a wooden bed, striped bedsheets, a lamp on the nightstand with its light turned on, and several large pillows arranged neatly from head to foot along the bed. The COCO-style caption should contain the most salient object and arrangement: the pillows.</think> Answer: A few pillows are arranged along the longer side of the bed.

---

Now answer for the following image: IMAGE: <image>

Figure 8: Prompt template for generating ECR data using teacher MLLMs for i2t tasks.

Given a caption describing an everyday scene, use your background knowledge and reasoning to expand the caption with more concrete and visual details. Follow the rules below:
- Generate a succinct reasoning process (2–3 sentences) about the visual information that can be inferred from the caption, using commonsense and visual priors.
- Wrap your reasoning in <think>...</think>. - Then output the final description of the corresponding image on the next line using: Answer: <answer>

---

EXAMPLE CAPTION: A teddy bear shop is equipped with a door guard teddy and a neighbor teddy above.

<think>The caption playfully describes a teddy bear shop, suggesting there are teddy bear figures used decoratively. A "door guard teddy" likely refers to a large bear figure placed at the shop entrance, possibly dressed in a costume to resemble a guard. The "neighbor teddy above" suggests another teddy is positioned at or hanging from an upper-story window, adding to the shop's whimsical appearance.</think> Answer: A teddy bear shop with a large teddy bear dressed as a royal guard standing at the entrance, and another teddy bear playfully hanging from an upstairs window.

---

Now answer for the following caption:
CAPTION: {query}

---

Given a caption, the goal is to describe what the target image may look like based on the caption and your background knowledge. You are given the target image for your reference. However,you should only use the provided caption and your background knowledge to reasonably infer what the image may depict.
Target image: <image>\n\n
Instructions:
- Use only the caption and general world knowledge to predict likely visual features.
- Do not make up details not implied by the caption. Be concise, yet specific and visual in your wording (1–2 sentences). Think of it as guiding someone to imagine the image — but only using what is grounded in the caption.
- Generate a succinct reasoning process (2–3 sentences) about the visual information that can be inferred from the caption, using commonsense and visual priors.
- Wrap your reasoning in <think>...</think>. - Then output the final description of the corresponding image on the next line using: Answer: <answer>
---
EXAMPLE
CAPTION: A man is riding a surfboard on a wave.
<think>A man wearing swimwear is balancing on a surfboard amid ocean waves, likely in a beach or sea setting, with water splashing around him.</think>
---
Now answer for the following caption:
CAPTION: {query}

Figure 9: Prompt template for generating ECR data using teacher MLLMs for t2v tasks.

## C  PROMPT TEMPLATES AND ECR SAMPLES

We provide sample prompts for prompting the teacher MLLMs (e.g. InternVL3 78B) to generate ECR. Examples are shown in Fig. 8, Fig. 9 and Fig. 7. We also provide sample generated ECR in Tab. 12, Tab. 13 and Tab. 14.

| Dataset | Input Text | Input Image | Embedding-Centric Reasoning |
|---------|-----------|-------------|----------------------------|
| ImageNet-1K | Given an image, identify the main object category it belongs to. Let's think step by step. |  | \<think\>The image mainly shows a hand plane with a metallic body and wooden handles, a tool used for smoothing or shaping wood.\</think\>Answer: hand plane |
| N24News | Given an image and its associated news text, identify the main domain of the news. Let's think step by step.
News text: Danai Gurira and Andrew Holland in a theater at New York University, where they met in the Tisch Graduate Acting Program. |  | \<think\>The image shows two individuals in a theater setting, and the caption mentions they are actors who met at a graduate acting program, which is related to the performing arts.\</think\>Answer: Arts & Entertainment |
| VOC2007 | Given an image, identify the main object category it belongs to. Let's think step by step. |  | \<think\>The image mainly shows a car with racing stripes and branding, identifying it as an automobile.\</think\>Answer: car |
| HatefulMemes | *Given an image, determine if it contains hateful speech. Let's think step by step.* |  | \<think\>The image shows a cute squirrel on top of a tree trunk. The text on the image is: "you're a special kinda nuts aren't you?" The text uses humor and wordplay referencing nuts and squirrels, which is harmless. There's no indication of harm or offense intended.\</think\>Answer: No |

Table 12: Examples of ECRs for Classification datasets in MMEB V1 benchmark.

| Dataset | Input Text | Input Image | ECR |
|---------|-----------|-------------|-----|
| A-OKVQA | Given the image and the below question, answer the question based on the image. Let's think step by step.
Question: What is the man by the bags awaiting? |  | \<think\>The man is standing by bags with a luggage cart on a street, a common scenario for someone waiting for transportation. The context suggests he is awaiting a cab.\</think\>Answer: cab |
| ChartQA | Given the image and the below question, answer the question based on the image. Let's think step by step.
Question: Is the value of Favorable 38 in 2015? |  | \<think\>The Favorable line is the green one. In 2015, the Favorable value is 38.\</think\>Answer: Yes |
| Visual7W | Given the image and the below question, answer the question based on the image. Let's think step by step.
Question: What is written on the white square on the bus? |  | \<think\>The image shows a double-decker bus with a large white square on its side that clearly displays the words "FOX'S GINGER BISCUITS" in bold, black letters.\</think\>Answer: FOX'S GINGER BISCUITS |

Table 13: Examples of ECRs for VQA datasets in MMEB V1 benchmark.

# D    FULL RESULTS

We present the full results on MMEB V2 (Jiang et al., 2024) in Tab. 15. For VLM2Vec 2B baseline, we use the officially reported number from VLM2Vec V2 (Meng et al., 2025). For VLM2Vec 7B, since it is not reported, we use our reproduced version under our training hyper-parameters.

| Dataset | Input Text | Input Image | ECR |
|---------|-----------|-------------|-----|
| VisualNews_i2t | Given a news image, first generate a concise and informative description of the news image, then generate a summarization based on the description. Let's think step by step. |  | \<think\>The image shows a man with a mustache wearing a dark suit and a purple tie with a pin or badge on his jacket. He appears to be in a formal setting, possibly outside a building. His expression is somber, suggesting a serious or official event. Based on his appearance and the context, this is likely a depiction of a prominent political figure at a significant moment.\</think\>Answer: Turkish President Recep Tayyip Erdogan is seen during a state visit or formal event. |
| WebQA | Given an wiki image and a short text description, first generate a detailed description of the image and text, then generate a summarization based on the description. Let's think step by step. Text: 1913-D Barber half obverse. |  | \<think\>The image shows a close-up of a silver Barber half obverse minted in 1913. The coin features a profile of Lady Liberty facing right, adorned with a laurel wreath and a ribbon tied behind the neck. Above the portrait, the words "IN GOD WE TRUST" are inscribed, and thirteen stars encircle the top and left side of Lady Liberty. The year "1913" is prominently displayed below the neck. The design is intricate, with fine details visible on the hair and leaves.\</think\>Answer: The image shows a 1913-D Barber half obverse featuring Lady Liberty with a laurel wreath and the inscription "IN GOD WE TRUST." |
| MSCOCO_t2i | Given a COCO-style caption, first use your knowledge to expand it into a more detailed and concise description of the target image, then generate a summarization based on the description. Let's think step by step. Caption: A skateboarder in mid air following a jump form cincrete. | | \<think\>The caption describes a skateboarder who has launched off a concrete surface. Given the mid-air position, the skateboarder is likely performing a trick such as an ollie or kickflip. The skateboarder is in the midst of executing the trick, and the concrete background suggests an urban skate park or street setting.\</think\>Answer: A skateboarder captured in mid-air, performing a daring trick off a concrete ramp in an urban skate park, surrounded by smooth pavement and scattered wheels |
| RefCOCO (Grounding) | Given an image and a query, identify the object or region in the image that the query refers to, and generate a concise description of the object or region. Let's think step by step. Query: vehicle second-closest to camera |  | \<think\>The image shows a black and white photograph of a small white dog. The dog is standing or walking on a light-colored surface, surrounded by a darker background. It has a round, fluffy appearance and is wearing a collar. The dog's features are somewhat blurred, making it difficult to discern fine details beyond its general shape and posture.\</think\>Answer: A small white dog standing on a light surface. |

Table 14: Examples of ECRs for Retrieval datasets in MMEB V1 benchmark.

| | ColPali v1.3 | GME-2B | GME-7B | LamRA Qwen2.5 | VLM2Vec 2B | VLM2Vec 7B | TTE$_s$-2B | TTE$_t$-2B | TTE$_s$-7B | TTE$_t$-7B |
|---|---|---|---|---|---|---|---|---|---|---|
| Avg - All (78 tasks) | 44.4 | 54.1 | 57.8 | 47.4 | 61.2 | 58.0 | 63.1 | 68.6 | 68.6 | 71.5 |
| Avg - Image (36 tasks, Hit@1) | 34.9 | 51.9 | 56.0 | 52.4 | 68.1 | 64.9 | 70.1 | 76.1 | 74.2 | 77.8 |
| Avg - Video (18 tasks, Hit@1) | 28.2 | 33.6 | 38.4 | 33.6 | 36.4 | 34.6 | 41.3 | 48.8 | 46.8 | 51.9 |
| Avg - Visdoc (24 tasks, NDCG@5) | 71.0 | 72.7 | 75.2 | 50.2 | 69.3 | 65.4 | 68.8 | 72.1 | 76.4 | 76.8 |
| I-CLS (10) | 40.3 | 54.4 | 57.7 | 51.7 | 65.7 | 62.9 | 67.9 | 76.6 | 69.7 | 76.7 |
| I-QA (10) | 11.5 | 29.9 | 34.7 | 34.1 | 61.5 | 56.3 | 66.6 | 76.8 | 72.4 | 78.6 |
| I-RET (12) | 48.1 | 66.9 | 71.2 | 66.9 | 70.0 | 69.5 | 70.2 | 71.5 | 71.5 | 74.3 |
| I-VG (4) | 40.3 | 55.5 | 59.3 | 56.7 | 85.2 | 77.3 | 84.1 | 87.2 | 90.6 | 89.3 |
| V-CLS (5) | 26.7 | 34.9 | 37.4 | 32.9 | 45.9 | 39.3 | 47.3 | 56.1 | 49.1 | 57.5 |
| V-QA (5) | 37.8 | 42.0 | 50.4 | 42.6 | 33.9 | 34.3 | 49.1 | 65.3 | 60.6 | 68.2 |
| V-RET (5) | 21.6 | 25.6 | 28.4 | 23.2 | 27.6 | 28.8 | 33.2 | 34.1 | 36.4 | 37.6 |
| V-MR (3) | 25.5 | 31.1 | 37.0 | 37.2 | 39.3 | 36.8 | 32.1 | 33.8 | 37.2 | 39.3 |
| VD-Vidore-V1 (10) | 83.6 | 86.1 | 89.4 | 56.3 | 78.8 | 75.5 | 77.5 | 81.1 | 84.1 | 83.7 |
| VD-Vidore-V2 (4) | 52.0 | 54.0 | 55.6 | 33.3 | 52.6 | 44.9 | 53.2 | 59.9 | 62.7 | 63.6 |
| VD-VisRAG (6) | 81.1 | 82.5 | 85.0 | 58.2 | 82.7 | 79.4 | 83.2 | 84.7 | 91.9 | 91.4 |
| VD-OOD (4) | 43.1 | 43.1 | 44.4 | 40.1 | 42.1 | 39.4 | 41.1 | 43.2 | 47.6 | 50.6 |
| ImageNet-1K | 42.4 | 58.3 | 64.6 | 58.9 | 82.5 | 80.8 | 83.3 | 83.1 | 84.3 | 84.6 |
| N24News | 25.5 | 50.1 | 50.5 | 29.8 | 80.1 | 72.9 | 78.6 | 83.1 | 83.1 | 81.8 |
| HatefulMemes | 50.6 | 52.5 | 53.6 | 51.3 | 67.9 | 56.3 | 64.0 | 78.2 | 67.4 | 75.8 |
| VOC2007 | 69.8 | 75.9 | 80.3 | 78.7 | 84.2 | 85.0 | 86.3 | 87.6 | 86.6 | 84.8 |
| SUN397 | 56.1 | 67.3 | 69.5 | 66.5 | 73.0 | 71.0 | 77.5 | 78.0 | 78.9 | 79.3 |
| Place365 | 27.5 | 35.8 | 39.1 | 37.4 | 41.7 | 35.9 | 45.7 | 59.8 | 44.6 | 64.1 |
| ImageNet-A | 14.9 | 28.8 | 41.2 | 36.3 | 49.6 | 47.4 | 50.9 | 69.8 | 60.4 | 73.0 |
| ImageNet-R | 64.6 | 78.6 | 83.9 | 77.0 | 88.4 | 89.3 | 89.7 | 90.2 | 90.5 | 90.5 |
| ObjectNet | 45.6 | 70.6 | 69.0 | 59.4 | 66.3 | 65.2 | 74.1 | 74.4 | 72.6 | 72.7 |
| Country211 | 6.0 | 26.5 | 24.8 | 21.7 | 23.7 | 25.2 | 28.5 | 62.0 | 29.0 | 60.3 |
| OK-VQA | 9.4 | 29.9 | 33.2 | 39.9 | 57.3 | 51.5 | 68.4 | 80.1 | 74.7 | 83.2 |
| A-OKVQA | 6.6 | 18.6 | 21.0 | 34.1 | 50.2 | 43.6 | 57.1 | 75.0 | 66.1 | 76.9 |
| DocVQA | 11.3 | 29.8 | 41.4 | 37.1 | 93.5 | 90.1 | 94.2 | 94.4 | 95.6 | 95.6 |
| InfographicsVQA | 5.0 | 11.6 | 20.3 | 23.7 | 69.3 | 58.8 | 65.6 | 81.8 | 77.5 | 81.6 |
| ChartQA | 5.7 | 13.4 | 17.8 | 15.0 | 56.8 | 47.4 | 57.5 | 81.7 | 70.9 | 83.1 |
| Visual7W | 6.1 | 16.2 | 22.2 | 24.6 | 55.3 | 52.9 | 54.1 | 70.7 | 57.9 | 73.3 |
| ScienceQA | 16.3 | 27.3 | 28.0 | 31.3 | 46.4 | 38.2 | 50.7 | 64.7 | 60.0 | 67.8 |
| VizWiz | 27.6 | 37.0 | 39.0 | 32.0 | 44.5 | 43.3 | 55.1 | 55.5 | 53.8 | 55.4 |
| GQA | 8.3 | 75.1 | 76.9 | 57.4 | 64.5 | 64.9 | 77.0 | 77.1 | 80.9 | 81.3 |
| TextVQA | 18.8 | 39.7 | 46.8 | 46.1 | 77.0 | 72.2 | 86.2 | 87.3 | 87.0 | 87.5 |
| VisDial | 41.2 | 48.1 | 60.8 | 62.5 | 82.3 | 82.7 | 81.2 | 81.5 | 84.4 | 84.9 |
| CIRR | 8.2 | 44.2 | 54.9 | 44.7 | 61.1 | 57.5 | 59.4 | 64.2 | 65.1 | 67.1 |
| VisualNews_t2i | 50.1 | 74.7 | 79.7 | 70.1 | 73.2 | 74.5 | 72.8 | 74.9 | 78.5 | 79.4 |
| VisualNews_i2t | 47.6 | 78.3 | 83.6 | 74.2 | 80.4 | 78.2 | 76.5 | 76.6 | 81.3 | 81.7 |
| MSCOCO_t2i | 59.2 | 68.1 | 71.2 | 65.7 | 75.8 | 75.3 | 75.2 | 75.7 | 77.9 | 78.4 |
| MSCOCO_i2t | 49.9 | 63.1 | 57.7 | 71.1 | 72.6 | 71.4 | 71.1 | 72.2 | 73.1 | 73.1 |
| NIGHTS | 65.5 | 67.0 | 67.6 | 64.4 | 66.5 | 68.6 | 70.8 | 68.0 | 69.8 | 70.2 |
| WebQA | 53.8 | 88.8 | 89.4 | 85.7 | 90.1 | 90.6 | 90.4 | 90.4 | 90.8 | 91.1 |
| FashionIQ | 5.9 | 32.9 | 37.8 | 33.4 | 24.9 | 19.5 | 26.3 | 28.9 | 29.7 | 30.2 |
| Wiki-SS-NQ | 80.5 | 73.9 | 78.2 | 67.0 | 72.2 | 66.9 | 64.2 | 64.9 | 70.5 | 71.2 |
| OVEN | 50.0 | 72.3 | 75.1 | 84.8 | 67.0 | 64.3 | 67.6 | 67.9 | 72.7 | 69.4 |
| EDIS | 64.7 | 91.8 | 96.0 | 78.7 | 73.6 | 84.1 | 87.0 | 92.8 | 93.9 | 95.2 |
| MSCOCO | 36.7 | 28.6 | 31.4 | 36.0 | 73.0 | 67.1 | 67.7 | 82.5 | 74.1 | 84.9 |
| RefCOCO | 64.5 | 55.9 | 60.9 | 57.1 | 93.1 | 87.1 | 91.4 | 89.4 | 97.7 | 92.1 |
| RefCOCO-Matching | 3.9 | 73.3 | 78.4 | 82.6 | 93.1 | 85.8 | 95.0 | 90.3 | 96.3 | 91.6 |
| Visual7W-Pointing | 56.1 | 64.1 | 66.5 | 51.2 | 81.7 | 69.2 | 82.5 | 86.5 | 94.3 | 88.4 |
| K700 | 23.4 | 35.2 | 39.7 | 32.1 | 53.6 | 38.0 | 49.6 | 48.2 | 55.0 | 55.6 |
| SmthSmthV2 | 25.1 | 29.9 | 30.6 | 25.3 | 46.6 | 42.8 | 50.4 | 59.4 | 44.9 | 55.3 |
| HMDB51 | 24.8 | 43.4 | 47.4 | 33.8 | 43.2 | 40.9 | 52.5 | 64.2 | 51.7 | 63.9 |
| UCF101 | 49.4 | 52.4 | 54.7 | 53.0 | 66.9 | 60.0 | 58.3 | 75.5 | 64.2 | 78.6 |
| Breakfast | 10.9 | 13.6 | 14.3 | 20.1 | 19.4 | 14.8 | 25.4 | 33.2 | 29.7 | 34.2 |
| MVBench | 33.7 | 37.5 | 46.6 | 37.6 | 32.8 | 33.7 | 48.5 | 62.0 | 59.5 | 65.3 |
| Video-MME | 30.6 | 34.3 | 39.2 | 35.1 | 33.1 | 30.7 | 45.8 | 58.9 | 53.1 | 62.1 |
| NExTQA | 35.2 | 39.5 | 46.8 | 44.9 | 21.0 | 20.9 | 53.8 | 74.0 | 70.1 | 73.6 |
| EgoSchema | 38.4 | 40.8 | 46.8 | 47.0 | 34.4 | 34.0 | 36.4 | 58.2 | 55.6 | 62.8 |
| ActivityNetQA | 51.3 | 58.0 | 65.6 | 48.5 | 48.0 | 52.3 | 60.8 | 73.6 | 64.6 | 77.1 |
| DiDeMo | 22.8 | 22.0 | 26.4 | 22.8 | 31.6 | 30.4 | 33.5 | 34.1 | 34.9 | 36.3 |
| MSR-VTT | 17.6 | 27.3 | 31.8 | 25.0 | 31.6 | 28.3 | 34.8 | 36.6 | 37.6 | 39.5 |
| MSVD | 45.4 | 47.6 | 49.7 | 41.9 | 46.7 | 48.1 | 56.5 | 57.1 | 58.5 | 59.4 |
| VATEX | 16.7 | 23.0 | 24.9 | 18.7 | 19.0 | 26.5 | 25.6 | 26.4 | 31.0 | 32.6 |
| YouCook2 | 5.3 | 7.9 | 9.4 | 7.5 | 9.2 | 10.6 | 15.8 | 16.3 | 19.9 | 20.3 |
| QVHighlight | 19.9 | 43.6 | 59.5 | 60.9 | 58.2 | 49.4 | 38.9 | 40.3 | 51.0 | 52.7 |
| Charades-STA | 29.0 | 14.9 | 14.0 | 18.8 | 19.3 | 20.2 | 19.5 | 21.4 | 18.9 | 23.0 |
| MomentSeeker | 27.6 | 34.8 | 37.4 | 31.8 | 40.6 | 40.8 | 37.7 | 39.6 | 41.5 | 42.2 |
| ViDoRe_arxivqa | 81.7 | 82.8 | 86.9 | 53.0 | 81.5 | 80.6 | 80.7 | 81.8 | 84.6 | 83.4 |
| ViDoRe_docvqa | 56.6 | 53.1 | 57.5 | 25.4 | 45.7 | 44.9 | 44.5 | 40.7 | 46.0 | 45.8 |
| ViDoRe_infovqa | 84.9 | 90.2 | 92.1 | 72.3 | 86.4 | 83.7 | 84.8 | 83.4 | 88.7 | 87.6 |
| ViDoRe_tabfquad | 86.9 | 93.3 | 94.6 | 66.1 | 91.4 | 89.2 | 88.4 | 90.1 | 94.7 | 92.0 |
| ViDoRe_tatdqa | 70.9 | 69.9 | 74.1 | 25.9 | 50.8 | 43.8 | 50.4 | 51.8 | 59.4 | 56.1 |
| ViDoRe_shiftproject | 75.1 | 89.5 | 96.8 | 27.3 | 71.8 | 60.8 | 65.2 | 79.3 | 81.6 | 81.8 |
| ViDoRe_artificial_intelligence | 95.7 | 97.5 | 99.6 | 72.0 | 91.0 | 88.5 | 91.9 | 97.3 | 98.1 | 98.2 |
| ViDoRe_energy | 94.7 | 91.9 | 95.3 | 65.2 | 86.8 | 86.5 | 88.7 | 93.4 | 93.5 | 96.2 |
| ViDoRe_government_reports | 93.6 | 94.6 | 98.8 | 72.2 | 87.8 | 85.0 | 86.9 | 95.3 | 96.7 | 97.7 |
| ViDoRe_healthcare_industry | 95.9 | 98.7 | 99.3 | 83.8 | 95.1 | 92.2 | 92.8 | 97.9 | 97.9 | 98.6 |
| ViDoRe_esg_reports_human_labeled_v2 | 51.3 | 61.0 | 63.4 | 33.0 | 55.7 | 45.6 | 59.0 | 64.9 | 69.4 | 70.9 |
| ViDoRe_biomedical_lectures_v2_multilingual | 54.7 | 54.0 | 49.5 | 35.9 | 54.0 | 44.3 | 52.0 | 55.0 | 60.8 | 62.8 |
| ViDoRe_economics_reports_v2_multilingual | 49.0 | 50.2 | 54.2 | 31.9 | 51.9 | 43.0 | 49.8 | 53.8 | 60.4 | 56.3 |
| ViDoRe_esg_reports_v2_multilingual | 52.9 | 50.7 | 55.4 | 32.5 | 48.8 | 46.6 | 52.1 | 65.8 | 60.3 | 64.3 |
| VisRAG_ArxivQA | 80.9 | 82.0 | 87.4 | 37.7 | 79.5 | 76.9 | 78.5 | 84.0 | 94.5 | 92.4 |
| VisRAG_ChartQA | 78.2 | 79.9 | 81.9 | 65.9 | 82.9 | 84.4 | 84.4 | 85.2 | 91.2 | 95.0 |
| VisRAG_MP-DocVQA | 86.8 | 84.4 | 89.2 | 54.5 | 81.5 | 71.8 | 79.2 | 80.0 | 90.1 | 87.0 |
| VisRAG_SlideVQA | 95.0 | 93.4 | 94.5 | 76.5 | 91.3 | 91.5 | 92.3 | 93.4 | 95.6 | 94.9 |
| VisRAG_InfoVQA | 85.7 | 91.4 | 93.5 | 73.3 | 89.8 | 85.7 | 87.2 | 87.8 | 93.0 | 92.2 |
| VisRAG_PlotQA | 60.3 | 64.1 | 63.4 | 41.2 | 71.5 | 66.1 | 77.5 | 77.8 | 86.9 | 87.1 |
| ViDoSeek-page | 22.2 | 21.6 | 23.2 | 23.1 | 21.7 | 21.9 | 22.6 | 22.9 | 35.0 | 45.2 |
| ViDoSeek-doc | 83.7 | 83.6 | 85.9 | 80.3 | 82.2 | 80.2 | 82.0 | 83.2 | 84.4 | 84.6 |
| MMLongBench-page | 14.2 | 15.8 | 16.2 | 13.5 | 14.8 | 11.9 | 12.9 | 17.8 | 20.7 | 22.0 |
| MMLongBench-doc | 52.5 | 51.4 | 54.3 | 43.5 | 49.7 | 43.7 | 47.0 | 48.8 | 50.4 | 50.6 |

Table 15: Full results on MMEB V2.

