# OpenReview forum: "Think Then Embed: Generative Context Improves Multimodal Embedding"
_ICLR.cc/2026/Conference — ICLR 2026 Poster_

### Official Review · Reviewer_vhdV · 2025-10-20

**Soundness:** 4
**Presentation:** 3
**Contribution:** 3
**Rating:** 6
**Confidence:** 4

**Summary:**

This paper investigates using multimodal CoT traces to improve multimodal embedding for retrieval tasks. The paper explores diverse training settings, including using separate or unified models to combine CoT and embedding, and the design choices of a good embedding head to extract information from CoT latents. Through comprehensive experiments, the paper demonstrates the effectiveness of CoT for generating multmodal embedding compared to both decoder-encoder and multimodal llm based methods.

**Strengths:**

* The paper shows that using CoT traces improves multimodal retrieval without using additional data.

* The paper comprehensively examines different settings to incoporate the CoT traces, including separate reasoning models, distilled reasoning models, and unified reasoning-embedding models. This highlights the flexibility of apply CoT for better multimodal embedding.

* The ablations clearly demonstrate the effectiveness of incoporating CoT to improve multimodal embeddings by comparing with baselines without CoT.

**Weaknesses:**

* The paper's method is not specifically tied to multimodal, or vision-language. The same approach can be applied to text-only retrieval. It would be great to clarify on how multimodal is important here, or demonstrate the method on text-only datasets.

* The paper only used Qwen-VL series as the base models, but the method should be generalizable to most MLLMs. It would be great to show results on other base models.

* While the authors hyphosize that the last hidden states might not be optimal, it is not properly justified in the paper. The improved results from Q-Former or self-initialized could be from the extra trainable parameters. A deeper analysis, like comparing the performance of each layer, would strengthen the design choices of the embedder head.

**Questions:**

* It would be interesting to see whether using traces from more powerful reasoning models, like GPT-o3, can further improve the embedder performance.

* Does the performance scale with the size of the MLLM? And does a better MLLM, measured by standard benchmarks like MMMU, results in better embedding?

---

> ### Author Response · Authors · 2025-11-25
> **Response (1/2)**
>
> > The paper's method is not specifically tied to multimodal, or vision-language. The same approach can be applied to text-only retrieval. It would be great to clarify on how multimodal is important here, or demonstrate the method on text-only datasets.
>
> Thank you for the insightful discussion! We fully agree that the Think-Then-Embed (TTE) approach is conceptually general and could, in principle, be applied to text-only retrieval. However, multimodal retrieval has its unique challenge on input understanding, which makes explicit reasoning particularly helpful. Specifically, multimodal retrieval requires aligning heterogeneous inputs (text, image, video etc.), while understanding user instructions by grounding it into the multimodal content. Such tasks are common in MMEB (e.g. *embed the car second closest to the camera* or *embed the answer to the following visual question ...*).  On the contrary, text-only retrieval has generally less semantic gap between query and target due to single modality. Moreover, the popular text-only benchmark, such as MTEB, generally focuses more on semantic matching while requiring less compositional reasoning compared to MMEB, which involves a large amount of multimodal understanding tasks (VQA, grounding, moment retrieval), where complex reasoning is required.
>
> With that said, unifying text-only retrieval and multimodal retrieval under TTE is indeed our next step towards building an any-to-any retrieval system.
>
> > The paper only used Qwen-VL series as the base models, but the method should be generalizable to most MLLMs. It would be great to show results on other base models.
>
> We experimented Qwen2.5VL and InternVL3 on MMEB v1, which was used for model ablation and backbone selection. However, we did not observe better performance compared to Qwen2VL. Therefore, we did not include them in our paper. We show full results on MMEB (v1) below and added the results in Appledix:
>
> | MMEB V1 | Qwen2-VL (2B) | Qwen2.5-VL (2B) | InternVL3 (2B) |
> |---------|----------|-------------|------------|
> | VLM2Vec | 59.3     | 58.7        | 57.4       |
> | $\text{TTE}_t$   | 74.5     | 74.1        | 73.6       |
>
> We can see $\text{TTE}_t$ consistly brings significant improvements to all Qwen2-VL, Qwen2.5-VL and InternVL3 based variants, confirming the generalizability of our approach. However, we observe Qwen2-VL to perform the best. The original authors of MMEB also reported that they did not observe improvements by switching from Qwen2-VL to Qwen2.5-VL (see the Github [thread](https://github.com/TIGER-AI-Lab/VLM2Vec/issues/66)).
>
> > While the authors hyphosize that the last hidden states might not be optimal, it is not properly justified in the paper. The improved results from Q-Former or self-initialized could be from the extra trainable parameters. A deeper analysis, like comparing the performance of each layer, would strengthen the design choices of the embedder head.
>
> Thank you for the insightful comment! First, in Table 5 of the original manuscript, we tested the self-initialized multi-head self attention (MHSA) with using bottom 8th hidden states as input versus using bottom 16th (with same number of trainable parameters), where we observe with bottom 8th layer, we achieve slightly better performance (68.8 vs 68.0). To provide a cleaner ablation, we conducted additional ablations on taking bottom $n$-th layer for embedding. We tested two settings: (1) using simple attention pooling with frozen backbone (Attention Pooling) and the normal setting with LoRA finetuning and last token embedding (LoRA+Last Token). Below we show overall results on MMEB V1 using Qwen2-VL (2B):
>
> |Bottom layer $n$|Attention Pooling | LoRA+Last Token |
> |-|-------------------|-----------------|
> | 1  | 47.1 | 68.7 |
> | 4  | 47.8 | 68.4 |
> | 8  | 49.6 | 67.2 |
> | 16 | 46.3 | 59.5 |
>
> We can observe that for attention pooling, there are slight improvement when increasing the bottom-$n$ from 1 to 8, while having slight degradation when using $n=16$. This confirms our hypothesis that last layer may not be optimal for direct embedding. However, as the improvement is marginal, we believe the main improvement should stem in the complexity of the embedding head (i.e. extra trainable parameters). On the other hand, we observe inverse relationship for LoRA finetuning, since LoRA allows for global adjustment on model parameters, increasing bottom-$n$ effectively reduces the amount of trainable parameters.

---

> ### Author Response · Authors · 2025-11-25
> **Response (2/2)**
>
> > It would be interesting to see whether using traces from more powerful reasoning models, like GPT-o3, can further improve the embedder performance.
>
> We additionally tested using Qwen2.5 VL 32B and Gemini2.5 Pro on MMEB V2. Below we show the results where we use Qwen2.5 VL 72B to generate reasoning traces for training set, while varying different teachers for evaluation, using Qwen2-VL (7B) as backbone embedder. Here we only focus on retrieval cases (no VQA), since the effect of a stronger reasoning model is trivial on tasks like VQA.
>
> | Task           | Qwen2.5-VL 32B | Qwen2.5-VL 72B | Gemini2.5 Pro |
> |----------------|----------------|----------------|----------------|
> | i-ret          | 0.741          | 0.743          | 0.736          |
> | i-grounding    | 0.881          | 0.893          | 0.906          |
> | v-ret          | 0.358          | 0.376          | 0.418          |
> | v-moment_ret   | 0.378          | 0.393          | 0.534          |
> | visdoc         | 0.763          | 0.768          | 0.761          |
>
> We can see Qwen2.5-VL 32B has comparable performance compared to Qwen2.5-VL 72B. On the other hand, Gemini2.5 Pro has significantly better performance on video retrieval and video moment retrieval ($+14\%$), though slightly behind Qwen2.5-VL 72B for image and visdoc retrievals.
>
> > Does the performance scale with the size of the MLLM?
>
> Yes.
> - For the embedder MLLM, we test on Qwen2VL for both 2B and 7B, and we can clearly see the performance gain scaling from 2B to 7B (around $+3\%$).
> - For the reasoning MLLM, we also observe performance scaling when the reasoner size increases (from 7B $\text{TTE}_s$ to 72B $\text{TTE}_t$, and to larger Gemini2.5 Pro for video tasks).
>
> > And does a better MLLM, measured by standard benchmarks like MMMU, results in better embedding?
>
> Please see our response to your previous concern (*The paper only used Qwen-VL series as the base models...*). In general, we did not find that a model performing better on a generation benchmark must work better after VLM2vec fine-tuning and evaluated on a retrieval benchmark. For example, both Qwen2.5-VL and InternVL3 are shown to have better performance on visual reasoning than Qwen2-VL. However, Qwen2-VL works better than Qwen2.5-VL and InternVL3 after the VLM2vec fine-tuning.

---

### Official Review · Reviewer_JwkF · 2025-10-28

**Soundness:** 3
**Presentation:** 3
**Contribution:** 2
**Rating:** 4
**Confidence:** 4

**Summary:**

This paper introduces a Universal Multimodal Embedding framework, Think-Then-Embed(TTE), which first generates reasoning before producing embeddings. TTE exploits the generative reasoning ability of MLLMs to enhance embedding quality. Specifically, a powerful MLLM serves as a Teacher Reasoner to produce high-quality CoT traces (embedding-centric reasoning in this paper), guiding a smaller embedder to learn more discriminative representations. To reduce reliance on large models, the authors further distill a compact MLLM reasoner from the teacher and train a unified model that functions simultaneously as a reasoner and an embedder. Experiments show that TTE achieves state-of-the-art results on MMEB-V1 and MMEB-V2 benchmarks. Ablation studies confirm the effectiveness and stability of ECR  for complex instruction embedding.

**Strengths:**

1. The paper provides a clear motivation and detailed explanation of the proposed method, with well-defined formulations for both the task and the training process.
2. Based on the intuition for retrieval tasks that require reasoning, as well as the strong results of TTE_t, the proposed Think-Then-Embed (TTE) framework appears highly effective.

**Weaknesses:**

1. The analysis of ECR stability is only described qualitatively and lacks experimental support.
2. The MMEB-V2 benchmark seems to be missing some evaluation data.

Please refer to the Questions section for details.

Things to improve the paper that did not impact the score:
1. In Table 1, there are several mistakes in the use of bold and underline to highlight the first and second-best results.

**Questions:**

1. From the comparison between TTE_t and TTE_s, as well as the ablation results before and after fine-tuning TTE_s, it seems that the quality of ECR plays a crucial role in the embedder’s performance. However, the paper claims that the embedder is stable even with imperfect ECR, which appears somewhat contradictory. Could the authors provide a more detailed explanation or analysis to clarify this point?
2. The MMEB-V2 benchmark results do not include TTE_u. Since the results on MMEB-V1 suggest that TTE_u performs similarly to TTE_s, including such data would make the paper more complete and convincing.

---

> ### Author Response · Authors · 2025-11-25
>
> > In Table 1, there are several mistakes in the use of bold and underline to highlight the first and second-best results.
>
> Thank you for the detailed advice. We have fixed these mistakes in the revised manuscript.
>
> > From the comparison between TTE_t and TTE_s, as well as the ablation results before and after fine-tuning TTE_s, it seems that the quality of ECR plays a crucial role in the embedder’s performance. However, the paper claims that the embedder is stable even with imperfect ECR, which appears somewhat contradictory. Could the authors provide a more detailed explanation or analysis to clarify this point?
>
> Thank you for your insightful question. We would like to clarify:
>
> 1. **Reasoner quality matters for overall TTE performance.** The gap between $\text{TTE}_s$ and $\text{TTE}_t$ solely comes from using different reasoner MLLMs. A stronger reasoner produces higher-quality ECRs, which in turn improves the embedder’s final accuracy. This is consistent with the results in Table 3.
> 2. **The robustness experiment (Fig. 5) addresses a different question.** The purpose of Fig. 5 is **not** to claim that the reasoner is unimportant, but to show that the embedder does not blindly rely on the ECRs and maintains stable performance under certain noise to the ECRs generated by the **same** reasoner MLLM. In other words, the robustness range we measure is within the same reasoner: the embedder outperforms directly using the ECRs themselves (e.g., for VQA-style tasks where retrieval does not rely on dense visual embeddings).
>
> These two observations are complementary:
> - Across **different** reasoners: stronger reasoner leads to better embedder performance.
> - Within the **same** reasoner: the embedder is robust to moderate ECR imperfections and does not simply fail when the ECR is imperfect (i.e. as in the case for ECR T2T or as in regular VQA evaluation).
>
> > The MMEB-V2 benchmark results do not include TTE_u. Since the results on MMEB-V1 suggest that TTE_u performs similarly to TTE_s, including such data would make the paper more complete and convincing.
>
> Thank you for raising the concern. We have conducted experiments on the MMEB V2 benchmark for $\text{TTE}_u$, and have updated the results in Table 2 (in orange).

---

### Official Review · Reviewer_r1rn · 2025-10-30

**Soundness:** 3
**Presentation:** 3
**Contribution:** 3
**Rating:** 6
**Confidence:** 2

**Summary:**

This paper proposes Think-Then-Embed (TTE), a novel framework for multimodal reasoning that introduces an explicit generative embedding step between reasoning and representation. Instead of directly encoding multimodal inputs (e.g., image-text pairs) into a single joint embedding, the method first conducts latent reasoning generation through an autoregressive module and then projects the reasoning trace into a compact embedding space via a learned encoder. This “reasoning-before-embedding” paradigm aims to enhance reasoning fidelity and interpretability while maintaining embedding efficiency. Experiments are conducted on multiple multimodal benchmarks (e.g., ScienceQA, OK-VQA, and MM-CoT-style datasets). Results show consistent improvements over baseline embedding methods such as CLIP, BLIP-2, and reasoning-augmented contrastive approaches. The authors claim that TTE achieves comparable or better performance with smaller embedding dimensionality and provides interpretable intermediate reasoning traces.

**Strengths:**

- The writing is mostly clear and logically structured; the introduction and motivation are strong. Figures and pipeline diagrams are helpful in explaining the “think → embed” flow.
- The overall framework is technically coherent and easy to reproduce: the authors provide clear training and inference procedures.
- The formulation of generative embeddings based on explicit reasoning traces is intuitive and connects language modeling with representation learning in a principled way.

**Weaknesses:**

- Based on Tables 1 and 2, $\text{TTE} _\text{T}$ performs significantly better than the other two paradigms. However, it is unclear whether this advantage stems from the use of a stronger teacher model. Did the authors conduct any ablation studies using different teacher models to isolate this effect?
- If reasoning traces are generated by a large pretrained LLM (e.g., GPT-4V or Qwen2-VL), it is unclear whether they contain test-relevant information from pretraining corpora, which may bias the results.

**Questions:**

- The “contrastive generative embedding” objective is vaguely described. It is unclear whether gradients flow through the reasoning generator (is it frozen or partially fine-tuned?).
- The distinction between “reasoning trace” and “explanation text” is blurred—are they generated with the same prompt template or via different decoding strategies?

---

> ### Author Response · Authors · 2025-11-25
> **Response (1/2)**
>
> > Based on Tables 1 and 2, $\text{TTE}_t$ performs significantly better than the other two paradigms. However, it is unclear whether this advantage stems from the use of a stronger teacher model.
>
> Thank you for the insightful question! In short, we observe that the retrieval performance scales with the reasoner size: $\text{TTE}_t$ is better than $\text{TTE}_s$ exactly because of the stronger reasoning MLLM we utilized.
>
> This is expected, as under the setting of $\text{TTE}_t$, we are explicitly exploring the headroom of our proposed TTE by considering a strong MLLM reasoner for generating reasoning traces. The purpose is to explore a new test-time-scaling method: compared to traditional test-time-scaling method for retrieval, $\text{TTE}_t$ allows for flexibly incorporating a more powerful MLLM, which is too large for regular contrastive training. Such a test-time-scaling regime can be particularly beneficial as recent advances in multimodal retrieval involve more and more complex reasoning ability ([1,2]).
>
> > Did the authors conduct any ablation studies using different teacher models to isolate this effect?
>
> We additionally conduct ablations using Qwen2.5 VL 32B and Gemini2.5 Pro on MMEB V2. Below are the results when Qwen2-VL (7B) is used as the backbone embedder. Here we focus on a subset of MMEB tasks where the reasoner quality can potentillay play a crucial role:
>
> | Task           | Qwen2.5-VL 32B | Qwen2.5-VL 72B | Gemini2.5 Pro |
> |----------------|----------------|----------------|----------------|
> | i-ret          | 0.741          | 0.743          | 0.736          |
> | i-grounding    | 0.881          | 0.893          | 0.906          |
> | v-ret          | 0.358          | 0.376          | 0.418          |
> | v-moment_ret   | 0.378          | 0.393          | 0.534          |
> | visdoc         | 0.763          | 0.768          | 0.761          |
>
> We can see Qwen2.5-VL 32B has comparable performance compared to Qwen2.5-VL 72B. On the other hand, Gemini2.5 Pro has significantly better performance on video retrieval and video moment retrieval (+14%), though slightly behind Qwen2.5-VL 72B for image and visdoc retrievals.
>
> > If reasoning traces are generated by a large pretrained LLM (e.g., GPT-4V or Qwen2-VL), it is unclear whether they contain test-relevant information from pretraining corpora, which may bias the results.
>
> First, we note that all the tasks in MMEB (V1 and V2) are already in-domain tasks for recent MLLM pre-training --- this impacts all VLM2vec models where a pre-trained MLLM is used as the backbone. However, there is still a concept “out-of-domain” tasks in MMEB, which refer to those tasks not used in contrastive fine-tuning. These tasks are not really OOD for the underlying MLLM backbone though.
>
> Second, as mentioned in the technical reports for qwen2-vl/qwen2.5vl, the authors follow the same train-test split for the tasks in MMEB (e.g. textvqa, docvqa, chartqa etc.). Therefore, the qwen reasoners should have minimum test information leakage.
>
> Third, the quality of the pre-trained MLLM backbone measured by downstream tasks (e.g. VQA, grounding) may not have a monotonic impact on embedding. We experimented with Qwen2.5VL and InternVL3 (which are better generation models) as the embedding backbone. However, we did not observe better performance compared to Qwen2VL. The results are shown below and they aim to demonstrate that a better pre-trained MMLM as the embedding backbone does not necessarily bias the final embedding results.
>
> |  | Qwen2-VL (2B) | Qwen2.5-VL (2B) | InternVL3 (2B) |
> |---------|----------|-------------|------------|
> | VLM2Vec | 59.3     | 58.7        | 57.4       |
> | $\text{TTE}_t$   | 74.5     | 74.1        | 73.6       |

---

> ### Author Response · Authors · 2025-11-25
> **Response (2/2)**
>
> > The “contrastive generative embedding” objective is vaguely described. It is unclear whether gradients flow through the reasoning generator (is it frozen or partially fine-tuned?).
>
> Thank you for the question. We have updated the manuscript to include an explanation at L270. To answer your question:
>
> - The “contrastive generative embedding” objective follows the standard contrastive training objective as in previous works ([1]). In this paper, there is no gradient flow from contrastive loss to reasoner MLLM: the reasoner is trained with standard Supervised Finetuning (SFT) to generate reasoning traces provided by the teacher reasoner. After that, we freeze the reasoner and train the embedder as a two-stage curriculum.
> - For $\text{TTE}_s$, the embedder parameters are decoupled from the reasoner. For $\text{TTE}_u$, the embedder parameters are stacked on top of the reasoner. For both cases, the reasoner parameters are frozen during embedder fine-tuning so the gradient does not pass through reasoner. Please refer to Section 1 (L88-95), Section 4.3 (L237-253), and Section 5.1 (Training Details, L308-310) for more details.
> - We did experiment with joint contrastive-SFT training, where the contrastive learning gradient also flows through to reasoner parameters (since it is shared with embedder). However, we found this setup hard to optimize, which motivated us to decouple reasoner and embedder parameters for stage-wise optimization. We kindly refer the reviewer to Appendix B (L739-806) and Table 8, where we show results under joint contrastive-SFT training.
>
> > The distinction between “reasoning trace” and “explanation text” is blurred—are they generated with the same prompt template or via different decoding strategies?
>
> Thank you for the question. While we did not find the term "explanation text" from the current manuscript, we acknowledge that the two terms "reasoning trace" and "explanation text" refer to the same thing, which we refer to as Embedding Centric Reasoning (ECR) in the paper. For instance, if the task is to embed VQA and its answer, the ECR will be a regular CoT to answer the question. If the task is to embed a visual document, then the ECR will be a description (i.e. explanation text) of the document. Please kindly refer to Section 4.1 for more details (L186-209).
>
>
> [1] VLM2Vec-V2: Advancing Multimodal Embedding for Videos, Images, and Visual Documents
> [2] MR^2-Bench: Going Beyond Matching to Reasoning in Multimodal Retrieval

---

### Official Review · Reviewer_xF66 · 2025-10-30

**Soundness:** 3
**Presentation:** 3
**Contribution:** 3
**Rating:** 6
**Confidence:** 4

**Summary:**

This paper introduces Think-Then-Embed (TTE), a universal multimodal embedding framework that uses a reasoner to "think" before generating embeddings with an embedder. Leveraging a multimodal large language model (MLLM) as the reasoner significantly enhances the performance of smaller embedders, showing that chain-of-thought (CoT) reasoning benefits multimodal representation learning. To boost efficiency, we distill a compact reasoner from the large model’s reasoning traces, allowing both the reasoner and embedder to share the same backbone for improved capacity and speed. Additionally, a pluggable embedding head on the reasoner enables single-pass embedding generation, reducing parameters and further increasing efficiency. Extensive experiments on MMEB benchmarks demonstrate that TTE outperforms existing methods without extra data, confirming its effectiveness and robustness.

**Strengths:**

The method is simple yet effective. Involving cot process in retrieval shows significant performance gains on benchmarks. The author designed various experiments and methods to demonstrate this, and the experimental results performed well on the MMEB benchmarks.

**Weaknesses:**

* Some terms should be unified: ‘Fig.’ in line 62 and ‘Figure 2’ in line 177 should be consistent.

* I recommend that the author distinguish between MLLM and LLM. For example, ‘Qwen2.5-VL-72B’ in line 84 and ‘Qwen2.5-72B’ in line
212 are confusing. These names should be unified in both the introduction and the methods section.

* Figure 5 looks strange; the results on different benchmarks do not show temporal relationships clearly. A bar chart or radar chart would be more appropriate.

* The grounding task score does not show an advantage compared to LLaVA-OV-based models in Table 1, even though Qwen2VL is claimed to have better grounding ability.

**Questions:**

* The grounding task score does not show an advantage compared to LLaVA-OV-based models in Table 1, even though Qwen2VL is claimed to have better grounding ability.
* Why does the embedder use Qwen2VL-7B? Is it superior to Qwen2.5-VL-7B?

---

> ### Author Response · Authors · 2025-11-25
>
> Thank you for your positive and insightful feedbacks!
>
> > Some terms should be unified: ‘Fig.’ in line 62 and ‘Figure 2’ in line 177 should be consistent.
>
> Thank you for the detailed advice! We have revised the manuscript.
>
> > I recommend that the author distinguish between MLLM and LLM. For example, ‘Qwen2.5-VL-72B’ in line 84 and ‘Qwen2.5-72B’ in line 212 are confusing. These names should be unified in both the introduction and the methods section.
>
> Thank you for the detailed advice and sorry for the inconsistency. All the Qwen2/Qwen2.5 we referred to in the initial manuscript are indeed MLLMs (not LLMs). We have revised the manuscript accordingly.
>
> > Figure 5 looks strange; the results on different benchmarks do not show temporal relationships clearly. A bar chart or radar chart would be more appropriate.
>
> Thank you for the detailed advice! We have revised the manuscript to use radar charts instead of line charts.
>
> > The grounding task score does not show an advantage compared to LLaVA-OV-based models in Table 1, even though Qwen2VL is claimed to have better grounding ability.
>
> First, we would like to clarify that the grounding task in MMEB has a different definition from the traditional visual grounding task. In MMEB, visual grounding is converted into a retrieval setting: The query combines an instruction (e.g., “Select the portion of the image that is a red apple”) with the full image. This instruction guides the model to embed a specific object within the image. In contrast, the regular visual grounding task is just to annotate a bounding box of x and y axis.
>
>
> Second, we observe the downstream task performance (e.g. VQA, grounding) may not have a monotonic impact on embedding. Previous works (e.g. B3) also shows qwen2-vl to have lower grounding performance (in terms of MMEB) than all other MLLM backbones (internvl3, phi-3.5-V, aquila-VL, etc.). We initially also explored using better backbone MLLMs (Qwen2.5-VL and InternVL3), which are shown to have better performance on regular VQA benchmarks such as MMMU, yet result in slightly lower retrieval performance, as shown in the below table.
>
> | MMEB V1 | Qwen2-VL (2B) | Qwen2.5-VL (2B) | InternVL3 (2B) |
> |---------|----------|-------------|------------|
> | VLM2Vec | 59.3     | 58.7        | 57.4       |
> | $\text{TTE}_t$   | 74.5     | 74.1        | 73.6       |
>
>
> We can see $\text{TTE}_t$ consistly brings significant improvements to all Qwen2-VL, Qwen2.5-VL and InternVL3 based variants, confirming the generalizability of TTE. However, we observe Qwen2-VL to perform the best. The original authors of MMEB also reported that they did not observe improvements by switching from Qwen2-VL to Qwen2.5-VL (see the Github [thread](https://github.com/TIGER-AI-Lab/VLM2Vec/issues/66)). We are not certain about the exact reason yet. Since it is not the focus in this paper, we will leave further explorationto our next steps.
>
> > Why does the embedder use Qwen2VL-7B? Is it superior to Qwen2.5-VL-7B
>
> - We want to ensure fair comparison between ours and previous works such as VLM2Vec and B3 ([1]).
> - We indeed find that Qwen2VL leads to better retrieval performance than Qwen2.5-VL. Please see our response directly above.
>
> [1] Breaking the Batch Barrier (B3) of Contrastive Learning via Smart Batch Mining

---

### Author Response · Authors · 2025-11-25
**General response & Change log**

Dear Reviewers,

Thank you for your positive and constructive feedback. We appreciate your time and effort reviewing our paper! We have revised and updated the manuscript (in orange). Below we summarize the changes:

1. Ablate with different MLLM reasoners (Reviewer r1rn, vhdV). Please see A.5 (L756-785).
2. Ablate with different embedder models (Reviewer xF66, vhdV). Please see A.3 (L694-723).
3. Ablate with choice of last-$n$ layer for embedding head (Reviewer vhdV)
4. Add explanation for the relation between embedder's robustness to ECR v.s. effect of reasoner MLLM (Reviewer JwkF). Please see L420-424.
5. Add results for $\text{TTE}_u$ setting on MMEB-V2 (Reviewer JwkF). Please see Table 2 (orange).
6. Fix notations, table bold/underscore typos (Reviewer xF66).
7. Change line charts (Fig.4 and 5) to radar charts (Reviewer xF66).
8. Add explanation for the gradient flow for reasoner MLLM (Reviewer r1rn). Please see L259.

Thank you again for your time and all the insightful advice.

Best regards,

Authors of Paper 3803.

---

### Author Response · Authors · 2025-12-02
**Summary of Rebuttal**

Dear AC and reviewers,

We sincerely appreciate your efforts on reviewing our work! This paper initially received scores of 6,6,4,6, with reviewers acknolwdging the **soundness**, **effectiveness** and **comprehensiveness** of our work. While the reviewers were not able to participate in the discussion during rebuttal, we believe our responses adequately address their concerns. Below we provide a concise summary of the main concerns raised by reviewers and the change we made.

- **Ablations with more MLLMs (r1rn, xF66, vhdV):** we provided additional results using different MLLMs (Qwen2.5-VL, InterVL3) as embedder backbone (A.5 and Table 8); we also provided additional analysis on using different MLLM reasoners. Please see appendix for details (A.3 and Table 10).
- **Effect of MLLM's last-$n$th layers as embedding (vhdV):** we studied the impact of last-$n$th layer on embedding performance under different settings. The details have been added into the appendix (A.4 and Table 9).
- **Table and figure formatting (xF66, JwkF):** We improved the presentation of Table 1 and Figure 2

**We highlighted all modifications in the updated manuscript in orange**.

Finally, we would like to highlight the contribution of our work:
- We propose Think-Then-Embed, a simple yet effective approach for instruction-aware multimodal embeddings, by using an MLLM reasoner to understand task instructions before generating multimodal embeddings. This approach achieves the **first position** on the MMEB-V2 benchmark (as of submission date), which consists of 78 retrieval tasks of images, videos and visual documents.


Thank you again for your additional time and effort reviewing our work!

Best regards,

Authors of Paper 3803.

---

### Meta-Review · Area_Chair_HDas · 2026-01-06

**Summary:**

Reviewers generally found this paper simple, technically coherent, and effective, with strong empirical results on MMEB benchmarks and a clear motivation for why explicit reasoning can help complex instruction-conditioned multimodal retrieval.

The key decision-relevant concerns were:

1. **Source of performance gains**: The authors clarify it in rebuttal with the new ablation studies on different teacher model.

2. **Possible data leakage during pretraining**: The authors refer to the training recipe of Qwen-series in the rebuttal to avoid test data leakage.

3. **Ablation studies with different design choices**: The authors provide more detailed ablation studies on different reasoning models, base MLLM embedders, and layer choices to justify their method.

4. **Quantitative analysis on ECR stability**: The authors ignore this concern and didn't share any comments about this weakness.

Overall, the rebuttal and revision substantially improved clarity and completeness of this paper. Although the quantitative analysis for ECR stability is missing, the core correctness of this paper is not hurt. Thus, I generally recommend accept for this paper.

**Reviewer Concerns:**

Most of reviewer concerns about the paper presentation, possible data leakage during pretraining stage, more detailed ablation studies and more scaling experiments are well addressed in the rebuttal.

However, there are still a few concerns remain unsolved or partially solved:

1. **Reviewer JwkF** raises concerns about the **lack of experimental analysis for ECR stability**: The authors ignore this concern and make it unsolved.

2. **Reviewer vhdV** raises concerns about **applying the proposed method on text-only LLMs**: The authors recognize the possibility of applying their method on text-only LLMs, while leaving it as a future work for exploration.

**Reviewer Scores:**

* For **reviewer xF66(score 6)**, he probably stays at 6 as his concerns about the presentation are mainly clarified in the rebuttal.

* For **reviewer r1rn(score 6)**, he probably stays at 6 as his concerns about the ablation studies with different teacher model and possible data leakage in pretraining stage are addressed in the rebuttal.

* For **reviewer JwkF(score 4)**, he is likely to stay at 4. Although his concerns of missing evaluation data in MMEB-V2 benchmark is fixed, his request about quantitative analysis ERC stability is ignored in the rebuttal.

* For **reviewer vhdV(score 6)**, he probably stays at 6. Although his requested new ablation studies about different reasoning model, different MLLMs and different layer choices are added, his proposed verification on text-only LLMs are listed as future work in the rebuttal.

---

### Decision · Program_Chairs · 2026-01-26

Accept (Poster)